# SMART: Self-supervised Multi-task pretrAining with contRol Transformers

**Yanchao Sun**[†]**, Shuang Ma**[‡]**, Ratnesh Madaan**[‡]**, Rogerio Bonatti**[‡]**,
Furong Huang**[†] **and Ashish Kapoor**[‡]

[†]University of Maryland, College Park. {`ycs,furongh`}`@umd.edu`
[‡]Microsoft Redmond, WA. {`shuama,ratnesh.madaan,rbonatti,akapoor`}`@microsoft.com`

## Abstract

Self-supervised pretraining has been extensively studied in language and vision domains, where a unified model can be easily adapted to various downstream tasks by pretraining representations without explicit labels. When it comes to sequential decision-making tasks, however, it is difficult to properly design such a pretraining approach that can cope with both high-dimensional perceptual information and the complexity of sequential control over long interaction horizons. The challenge becomes combinatorially more complex if we want to pretrain representations amenable to a large variety of tasks. To tackle this problem, in this work, we formulate a general pretraining-finetuning pipeline for sequential decision making, under which we propose a generic pretraining framework *Self-supervised Multi-task pretrAining with contRol Transformer (SMART)*. By systematically investigating pretraining regimes, we carefully design a Control Transformer (CT) coupled with a novel control-centric pretraining objective in a self-supervised manner. SMART encourages the representation to capture the common essential information relevant to short-term control and long-term control, which is transferrable across tasks. We show by extensive experiments in DeepMind Control Suite that SMART significantly improves the learning efficiency among seen and unseen downstream tasks and domains under different learning scenarios including Imitation Learning (IL) and Reinforcement Learning (RL). Benefiting from the proposed control-centric objective, SMART is resilient to distribution shift between pretraining and finetuning, and even works well with low-quality pretraining datasets that are randomly collected. Our codebase, pretrained models and datasets are provided at https://github.com/microsoft/smart.

## 1 Introduction

Self-supervised pretraining has been successful in a wide range of language and vision problems. Examples include BERT (Devlin et al., 2019), GPT (Brown et al., 2020), MoCo (He et al., 2020), and CLIP (Radford et al., 2021). These works demonstrate that one single pretrained model can be easily finetuned to perform many downstream tasks, resulting in a simple, effective, and data-efficient paradigm. When it comes to sequential decision making, however, it is not clear yet whether the successes of pretraining approaches can be easily replicated.

There are research efforts that investigate application of pretrained vision models to facilitate control tasks (Parisi et al., 2022; Radosavovic et al.). However, there are challenges unique to sequential decision making and beyond the considerations of existing vision and language pretraining. We highlight these challenges below: (1) *Data distribution shift:* Training data for decision making tasks is usually composed of trajectories generated under some specific behavior policies. As a result, data distributions during pretraining, downstream task finetuning and even during deployment can be drastically different, resulting in a suboptimal performance (Lee et al., 2021). (2) *Large discrepancy between tasks:* In contrast to language and vision where the underlying semantic information is often shared across tasks, decision making tasks span a large variety of task-specific configurations, transition functions, rewards, as well as action and state spaces. Consequently, it is hard to obtain a generic representation for multiple decision making tasks. (3) *Long-term reward maximization:* The general goal of sequential decision making is to learn a policy that maximizes long-term reward.

Thus, a good representation for downstream policy learning should capture information relevant for both immediate and long-term planning, which is usually hard in tasks with long horizons, partial observability and continuous control. (4) *Lack of supervision and high-quality data:* Success in representation learning often depends on the availability of high quality expert demonstrations and ground-truth rewards (Lee et al., 2022; Stooke et al., 2021). However, for most real-world sequential decision making tasks, high-quality data and/or supervisory signals are either non-existent or prohibitively expensive to obtain.

Under these challenges, we strive for pretrained representations for control tasks that are
(1) **Versatile** so as to handle a wide variety of downstream control tasks and variable downstream learning methods such as imitation and reinforcement learning (IL, RL) etc,
(2) **Generalizable** to unseen tasks and domains spanning multiple rewards and agent dynamics, and
(3) **Resilient** and robust to varying-quality pretraining data without supervision.

We propose a general pretraining framework named *Self-supervised Multi-task pretrAining with contRol Transformer (SMART)*, which aims to satisfy the above listed properties. We introduce *Control Transformer (CT)* which models state-action interactions from high-dimensional observations through causal attention mechanism. Different from the recent transformer-based models for sequential decision making Chen et al. (2021) which directly learn reward-based policies, CT is designed to learn reward-agnostic representations, which enables it as a unified model to fit different learning methods (e.g. IL and RL) and various tasks. Built upon CT, we propose a control-centric pretraining objective that consists of three terms: forward dynamics prediction, inverse dynamics prediction and random masked hindsight control. These terms focus on policy-independent transition probabilities, and encourage CT to capture dynamics information of both short-term and long-term temporal granularities. In contrast with prior pretrained vision models (Oord et al., 2018; Parisi et al., 2022) that primarily focus on learning object-centric semantics, SMART captures the essential control-relevant information which is empirically shown to be more suitable for interactive decision making. SMART produces superior performance than training from scratch and state-of-the-art (SOTA) pretraining approaches on a large variety of tasks under both IL and RL. Our main contributions are summarized as follows:

1. We propose SMART, a generic pretraining framework for multi-task sequential decision making.
2. We introduce the Control Transformer model and a control-centric pretraining objective to learn representation from offline interaction data, capturing both perceptual and dynamics information with multiple temporal granularities.
3. We conduct extensive experiments on DeepMind Control Suite (Tassa et al., 2018). By evaluating SMART on a large variety of tasks under both IL and RL regimes, SMART demonstrates its *versatile* usages for downstream applications. When adapting to unseen tasks and unseen domains, SMART shows superior *generalizability*. SMART can even produce compelling results when pretrained on low-quality data that is randomly collected, validating its *resilience* property.

## 2 RELATED WORKS

**Offline Pretraining of Representation for Control.** Many recent works investigate pretraining representations and finetuning policies for the same task. Yang & Nachum (2021) investigate several pretraining objectives on MuJoCo with vector state inputs. They find that many existing representation learning objectives fail to improve the downstream task, while contrastive self-prediction obtains the best results among all tested methods. Schwarzer et al. (2021) pretrain a convolutional encoder with a combination of several self-supervised objectives, achieving superior performance on the Atari 100K. However, these works just demonstrated the single-task pretraining scenario, it is not clear yet whether the methods can be extended to multi-task control. Stooke et al. (2021) propose ATC, a contrastive learning method with temporal augmentation. By pretraining an encoder on expert demonstrations from one or multiple tasks, ATC outperforms prior unsupervised representation learning methods in downstream online RL tasks, even in tasks unseen during pretraining.

**Pretrained Visual Representations for Control Tasks.** Recent studies reveals that visual representations pretrained on control-free datasets can be transferred to control tasks. Shah & Kumar (2021) show that that a ResNet encoder pretrained on ImageNet is effective for learning manipulation tasks. Some recent papers also show that encoders pretrained with control-free datasets can generalize well to RL settings (Nair et al., 2022; Seo et al., 2022; Parisi et al., 2022). However, the generalizability of the visual encoder can be task-dependent. Kadavath et al. (2021) point out that ResNet pretrained on ImageNet does not help in DMC (Tunyasuvunakool et al., 2020) environments.

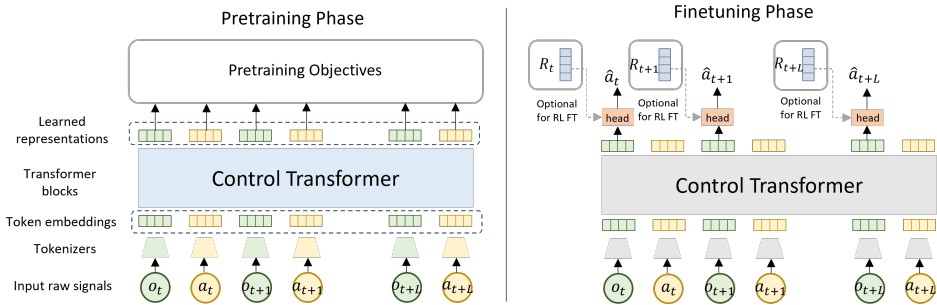

**Figure 1:** Architecture of Control Transformer. In the pretraining phase, we use the control-centric objective introduced in Section 5.2 to train representation over multiple tasks; in the finetuning phase where a specific task is given, we learn a policy based on the pretrained representation (pretrained weights are shown in grey blocks). The construction of the policy head can vary for different downstream datasets or learning methods.

**Unsupervised RL.** Unsupervised RL (URL) focuses on learning exploration policies (Pathak et al., 2017; Liu & Abbeel, 2021), goal-conditioned policies (Andrychowicz et al., 2017; Mendonca et al., 2021) or diverse skills (Eysenbach et al., 2019) in a task without external rewards, and finetuning the policy later when reward is specified. The unsupervised learning phase of URL is usually interactive and prolonged. Our goal, in contrast, is to train representations of states and actions from fixed offline datasets, with a focus on capturing essential and important information from raw inputs.

**Sequential Decision Making with Transformers.** There is a growing body of work that uses Transformer (Vaswani et al., 2017) architectures to model and learn sequential decision making problems. Chen et al. (2021) propose Decision Transformer (DT) for offline RL, which takes a sequence of returns, observations and actions, and outputs action predictions. Trajectory Transformer (Janner et al., 2021) also models the trajectory as a sequence of states, actions and rewards, while discretizing each dimension of state/actions. Bonatti et al. (2022) propose a pretraining scheme for state-action representations in navigation scenarios using a causal transformer, which can then be finetuned with imitation learning towards different tasks for the same robot. Furuta et al. (2022) propose Generalized DT that unifies a family of algorithms for future information matching with transformers. Zheng et al. (2022) extend DT to online learning by blending offline pretraining and online finetuning. Transformers can also be used as world models for model-based RL (Chen et al., 2022; Micheli et al., 2022). Recent studies show that transformer-based models can be scaled up with diverse multi-task datasets to produce generalist agents (Reed et al., 2022; Lee et al., 2022). Our proposed method has a similar structure that regards RL trajectories as sequential inputs. However, differently from most existing transformer models that learn policy from returns, our SMART focuses on learning control-relevant representations with self-supervised pretraining.

## 3 PRELIMINARIES

**Partially Observable Markov Decision Process.** We model control tasks and environments as a partially observable Markov decision process (POMDP) $\mathcal{M} = \langle \mathcal{S}, \mathcal{A}, \mathcal{O}, P, R, E \rangle$, which is a generalization of Markov decision process (MDP). Here, $\mathcal{S}$ is the underlying state space, $\mathcal{A}$ is the action space, $\mathcal{O}$ is the observation space, $P$ is the transition kernal, $R$ is the reward function, and $E$ is the observation emission function with $E(o|s)$ being the probability of observing $o$ given state $s$. In practice, the observation space can be high dimensional. For example, for a mobile robot navigating with camera sensors, the images are observations and its odometry (location, orientation, and associated velocities) and the ground-truth obstacle locations form the underlying state.

**Learning Agent and Controlling Policy.** At every step $t$, the agent receives an observation $o_t$ based on the underlying state $s_t$ (hidden from the agent), takes action $a_t$, and obtains a reward $r_t$ and the environment proceeds to the next state $s_{t+1}$. Given a history of observation-action pairs of length $L$ and the current observation, $h_t = (o_{t-L}, a_{t-L}, o_{t-L+1}, a_{t-L+1}, \cdots, o_t)$, the agent executes action $a_t$ according to policy $\pi$: $a_t = \pi(h_t)$. The agent's goal is to learn an optimal policy $\pi^*$ that maximizes the agent's cumulative reward $\mathbb{E}_P[\sum_{t=1}^{\infty} \gamma^t r_t]$.

**Reinforcement Learning (RL) and Imitation Learning (IL).** RL (Sutton & Barto, 2018) is the process where an agent seeks to maximize its policy returns. RL agents can learn in an online manner by interacting with the environment, or learn from offline data with pre-collected interactions. Differently from supervised learning, the training data for RL stems from policy-dependent inter-

actions with the environment, rendering a non-i.i.d. training regime. Another effective method for learning control policies is IL (Hussein et al., 2017; Osa et al., 2018), where an agent obtains supervision from expert demonstrations. For both RL and IL, there could be a discrepancy between training and deployment data distributions due to the different distribution of states induced by the imperfect policy and environment uncertainty.

## 4 PROBLEM SETUP: PRETRAINING AND FINETUNING PIPELINE

**Multi-Task Control with Shared Representation.** We consider a set of multiple tasks $\mathcal{T}$ with the same dimensionality in observation space. In this work we select $\mathcal{T}$ from different environment in DeepMind Control Suite (DMC) (Tassa et al., 2018), in which the agent observes an RGB image of the current state. Tasks in $\mathcal{T}$ can have entirely different state spaces $\mathcal{S}$, different action spaces $\mathcal{A}$ and different environment dynamics $(P, R, E)$. We also define the concept of *domain* to differentiate tasks that have different state/action spaces. For example, in DMC, "hopper" and "walker" belong to different domains because they posses distinct action spaces, while "walker-walk" and "walker-run" are different tasks within the same domain. In this paper we use the term *multi-task* to refer tasks spanning potentially multiple domains.

**Pretraining-Finetuning Pipeline.** Although pretraining methods are widely applied in many areas, it is not yet clear what role pretraining should play in sequential decision making tasks, especially when considering the multi-task setup. In this work, we follow ideas established in vision and language community to explicitly define our pretraining and finetuning

**Table 1:** A comparison between pretraining and finetuning.

| Pretraining phase | Finetuning phase |
|---|---|
| Learn generic representation | Learn policy |
| Offline | Offline or online |
| Multiple tasks | One task, seen or unseen |
| Reward or expert demonstration may be absent | Has reward supervision or expert demonstration |
| More samples | Fewer samples |

pipeline, which we summarize in Table 1. Specifically, during the pretraining phase we train representations with a possibly large offline dataset collected from a set of training tasks $\mathcal{T}_{\text{pre}} = \{\mathcal{M}_i\}_{i=1}^n$. Then, given a specific downstream task $\mathcal{M}$ which may or may not be contained in $\mathcal{T}_{\text{pre}}$, we attach a simple policy head on top of the pretrained representation[1] and train it with IL or with RL. The central tenet of pretraining is to learn generic representations which allow downstream task finetuning to be simple, effective and efficient, even under low-data regimes.

This pretraining-finetuning pipeline is a general extension of many prior settings of pretraining for decision making. For example, Stooke et al. (2021) pretrain an encoder on one or multiple tasks, then learn an RL policy in downstream tasks. Their pretraining dataset is composed of expert demonstration, and the finetuning process focuses on online RL. In addition, Schwarzer et al. (2021) learn representations with offline datasets, but perform pretraining and finetuning within the same task.

## 5 OUR PRETRAINING MODEL AND APPROACH

We propose Self-supervised Multi-task pretrAining with contRol Transformer (SMART), a general pretraining approach for multi-task sequential decision making. We first give an overview of the proposed Control Transformer (CT) and illustrate how it fits in our pretraining-finetuning pipeline in Section 5.1. Then, we introduce our control-centric pretraining objective in Section 5.2.

### 5.1 APPROACH OVERVIEW AND MODEL ARCHITECTURE

**Model Architecture of Control Transformer.** Inspired by the recent success of transformer models in sequential modeling (Chen et al., 2021; Janner et al., 2021), we propose a Control Transformer (CT). The input to the model is a control sequence of length $2L$ composed of observations and actions: $(o_t, a_t, o_{t+1}, a_{t+1}, \cdots, o_{t+L}, a_{t+L})$. Different from Decision Transformer (DT) (Chen et al., 2021), we purposefully do not include the reward signal in the control sequence to keep our representations reward-agnostic, as explained in the end of Section 5.1. Each element of the sequence is embedded into a $d$-dimensional token, with a modality-specific tokenizer jointly trained with Transformer blocks. We also learn an additional positional embedding and sum it with each token. The outputs of CT correspond to token embeddings representing each observation and action, and are represented by $\phi(o_t)$ and $\phi(a_t)$, respectively.[2] Figure 1 depicts the CT architecture.

---

[1]The pretrained encoder can be either frozen or finetuned with the policy, depending on the task.

[2]The implementation details are provided in Appendix A.3.

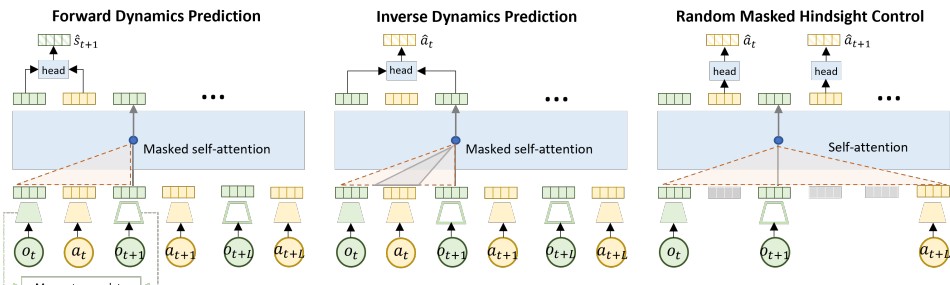

**Figure 2:** The three terms of our proposed pretraining objective. The red shaded areas denote the attention span, while the grey regions are masked.

**Pretraining of SMART.** We generate an offline dataset for pretraining which contains control trajectories generated by some behavior policies for a set of diverse tasks $\mathcal{T}_{\text{pre}}$ spanning multiple domains. During pretraining, we append several prediction heads to the transformer output, and train the entire model by minimizing the control-centric pretraining objective introduced in Section 5.2. These prediction heads are used to learn desired representations, will be dropped in finetuning.

**Downstream Finetuning of SMART.** As discussed in Section 4, the pretrained representation can be used to learn policies for different tasks. To do so, we append a policy head $\pi$ to the observation representation, such that $\pi(\phi(o_t))$ predicts the proper action for observation $o_t$. We can train the policy head using both IL and RL. For our IL experiments we use behavior cloning with expert demonstrations, where we feed $\phi(o_t)$ into a policy head to get action predictions. For RL we use a return-to-go (RTG)-conditioned policy with trajectories that contain reward values. We feed $\phi(o_t)$ along with an RTG embedding to get the policy head's action predictions. Online RL with transformer-based models is still a novel field and is not the focus of this work. But we show advantages of our pretraining method for online RL finetuning, following the same settings as online DT presented by Zheng et al. (2022). See Appendix B.4 for more discussion and results.

**Comparison with Prior Decision-making Transformers.** Recent works leverage transformer architectures for modeling sequential decision making problems (Chen et al., 2021; Janner et al., 2021; Lee et al., 2022) as summarized in Section 2. Most of these models use reward information in the input sequence, as their goal is to directly learn a policy for a specific task. In contrast, our goal is to pretrain representations for various downstream tasks, and thus our CT uses reward-free control sequence as the model input; rewards are only used for downstream task when needed. There are several **benefits** of making representation agnostic to reward during pretraining. **(1)** A pretrained model requiring reward input does not flexibly fit some downstream learning scenarios such as behavior cloning, while CT can be a unified model for various learning methods. A user can easily learn a policy under different learning methods (IL or RL) without modifying transformer blocks. **(2)** Reward distribution can be significantly different when the task or policy changes, making a reward-dependent representation less resilient to distribution shift. We show in Section 6.3 that *utilizing reward during pretraining may hurt the overall downstream performance*.

## 5.2 CONTROL-CENTRIC SELF-SUPERVISED PRETRAINING OBJECTIVES

Our pretraining objective employs three terms: forward dynamics prediction, inverse dynamics prediction, and random masked hindsight control. The first two terms focus on local and short-term dynamics, while the third term is designed to capture more global and long-term temporal dependence. As motivated in Section 5.1, these terms are based on control sequences and are reward-free, such that they can be used for multiple tasks. Figure 2 illustrates each objective.

**I. Forward Dynamics Prediction.** For each observation-action pair $(o_t, a_t)$ in a control sequence, we aim to predict the next immediate latent state. Let $g$ be the observation tokenizer being trained. We maintain a momentum encoder $\bar{g}$ as the exponential moving average of $g$, to generate the target latent state from the next observation $o_{t+1}$, i.e., $\hat{s}_{t+1} := \bar{g}(o_{t+1})$. The idea of momentum encoder is widely used when the target value is not fixed (He et al., 2020; Mnih et al., 2015), in order to make training more stable. Then, we train a linear head to predict $\hat{s}_{t+1}$ based on $(\phi(o_t), \phi(a_t))$. This forward prediction captures the local transition information in the embedding space.

**II. Inverse Dynamics Prediction.** For each consecutive observation pair $(o_t, o_{t+1})$, we learn to recover the action that leads $o_t$ to $o_{t+1}$ Note that in a causal transformer, $a_t$ is visible to the model

when generating representation of $o_{t+1}$ which can lead to trivial solutions, so we modify the original causal mask and mask out $a_t$ from the attention of $o_{t+1}$. Therefore, the observation representation is forced to contain information for relevant actions and transitions.

Both forward and inverse predictions focus on local dynamics induced by the transition kernel $P$. However, fitting local dynamics only may result in collapsed representation, i.e., the model learns the same representation for two semantically different observations (Rakelly et al., 2021). To perform well in downstream tasks, long-term temporal dependence should also be captured in the representation. We achieve this by a novel random masked hindsight control term in pretraining.

**III. Random Masked Hindsight Control.** Given a control sequence $h = (o_t, a_t, \cdots, o_{t+L}, a_{t+L})$, we randomly mask $k$ actions and $k'$ observations, and recover the masked actions based on the remaining incomplete sequence. This idea of masked token prediction is related to BERT (Devlin et al., 2019) for language modeling, but note that we only predict the masked actions for the purpose of control. More rationale behind the selection of $k$ and $k'$ is explained in Appendix A.1. This objective is akin to asking the question "what actions should I take to generate such a trajectory?" Therefore, we replace the causal attention mask with a non-causal one, to temporarily allow the model "see the future", as shown in Figure 2(right). As a result, we encourage the model to learn controllable representations and global temporal relations, and to attend to the most essential representations for multi-step control. The idea of our random masked hindsight control is also related to the multi-step inverse prediction proposed by a concurrent work (Lamb et al., 2022), which predicts $a_t$ given $s_t$ and $s_{t+l}$ for a random interger $l$ and theoretically shows the effectiveness of this method in discovering controllable states. Our random masked hindsight control is different as it predicts multiple actions altogether from randomly masked sequences with a transformer model, which can efficiently learn the control information in large-scale tasks, and avoid ambiguity caused by different paths between states. An empirical comparison between our formulation and the multi-step inverse prediction (Lamb et al., 2022) is provided in Appendix B.5.2.

Finally, our pretraining objective is the summation of the above three terms with equal weights, which in experiments renders good performance. The mathematical formulations and implementation details of the objective are explained in Appendix A.1.

## 6 EXPERIMENTS

We provide empirical results to demonstrate the effectiveness of our proposed pretraining method, while aiming to answer the following questions: **(1)** Can SMART effectively improve learning efficiency and performance in a variety of downstream tasks under different learning methods? **(2)** How well can SMART generalize to out-of-distribution tasks? **(3)** Is SMART resistant to low-quality pretraining data? **(4)** How does SMART compare to state-of-the-art pretraining techniques? **(5)** How do different pretraining objectives affect the downstream performance?

### 6.1 EXPERIMENTAL SETUP

We evaluate SMART on the DeepMind Control (DMC) suite (Tassa et al., 2018), which contains a series of continuous control tasks with RGB image observations. There are multiple domains (physical models with different state and action spaces) and multiple tasks (associated with a particular MDP) within each domain, which creates diverse scenarios for evaluating pretrained representations. Our experiments use 10 different tasks spanning over 6 domains. In pretraining, we use an offline dataset collected over 5 tasks, while the other 5 tasks (with 2 unseen domains) are held out to test the generalizability of SMART. A full list of tested domains and tasks is in Appendix A.2.

**Pretraining Tasks and Datasets.** We pretrain SMART on 5 tasks: cartpole-swingup, hopper-hop, cheetah-run, walker-stand and walker-run. For each task, we adopt/train behavior policies to collect the following two types of offline datasets. (Details of dataset collection are in Appendix A.2.)
- `Random`: Trajectories with random environment interactions, with 400K timesteps per task.
- `Exploratory`: Trajectories generated in the exploratory stage of multiple RL agents with different random seeds, with 400K timesteps per task.

**Downstream Tasks, Learning Methods and Datasets.** We evaluate the pretrained models in the 5 seen tasks and another 5 unseen tasks: cartpole-balance, hopper-stand, walker-walk, pendulum-swingup and finger-spin (the last two tasks are from unseen domains with different state-action spaces). We consider two learning methods: return-to-go-conditioned policy learning (RTG) and behavior cloning (BC). For RTG, we use the `Sampled Replay` dataset containing randomly sam-

pled trajectories from the full replay buffer collected by learning agent. For BC, we use the `Expert` trajectories with the highest returns from the full replay buffer of learning agents. The downstream dataset for every task only has 100K timesteps, making it challenging to learn from scratch.

**Implementation Details.** Our implementation of CT is based on a GPT model (Radford et al., 2018) with 8 layers and 8 attention heads. We use context length $L = 30$ and embedding size $d = 256$. As explained in Appendix A.1, $k$ and $k'$ are linearly increased from 1 to $L$ and $L/2$, respectively. The observation tokenizer is a standard 3-layer CNN. Action tokenizer, return tokenizer, and all single-layer linear prediction heads. We found that freezing the pretrained weights in downstream tasks works well in relatively simple environments, but fails in harder ones. Therefore, we finetune the entire model including transformer blocks for all downstream tasks. Since actions in different domains have different dimensions and physical meanings, we project the raw actions into a larger common action space to train the action tokenizer. When there is a novel downstream task with a different action space, we simply re-initialize the action tokenizer and finetune it. Please see Appendix A.3 for more implementation and hyperparameter details.

**Baselines.** We compare SMART with the following transformer-based pretraining baselines:

- `Scratch` trains a policy with randomly initialized CT representation weights.
- `ACL` (Yang & Nachum, 2021) is a modified BERT (Devlin et al., 2019) that randomly masks and predicts tokens with a contrastive loss, pretrained on the same dataset as SMART.
- `DT` (Chen et al., 2021) pretrained on the same dataset as ours but uses extra reward supervision.
- `CT-single` is a variant of SMART, which pretrains CT with a single-task dataset containing trajectories from the downstream environment.

For fair comparisons, we use the same network architecture for the baseline models (except for DT where we keep their original network structure with RTG as transformer inputs) and train them with the same configurations. We also compare SMART with other state-of-the-art pretraining works, such as `CPC` (Oord et al., 2018) and `ATC` (Stooke et al., 2021), using the same pretraining-finetuning pipeline. However, as these approaches are built upon ResNet backbones, a direct comparison of a Transformer against a ResNet could be not straightforward. Hence we refer readers to Appendix B.3 for more discussions and results.

**Evaluation Metrics.** To evaluate the quality of pretrained representations we report the average cumulative reward obtained in downstream tasks after finetuning. We deploy the trained policies in each environment for 50 episodes and report average returns. For evaluation of the RTG-conditioned policies, we use the expert score of each task as the initial RTG, as done in DT (Chen et al., 2021). Detailed evaluation settings as described in Appendix A.4.

## 6.2 EXPERIMENTAL RESULTS

We first evaluate the **versatility** of SMART: 1) whether a single pretrained model can be finetuned with different downstream learning methods (i.e. RTG and BC); and 2) whether the pretrained model can adapt towards various downstream tasks. Figure 3 compares the the reward curve of SMART with `Scratch` and `CT-Single`, where models are pretrained with `Exploratory` dataset[3]. To avoid overlapping of curves, comparison with `ACL` and `DT` is shown and discussed later in Figure 5. It can be seen that pretrained CT from both single-task dataset (`CT-single`) and multi-task dataset (SMART) can achieve much better results than training from scratch. In general, under both RTG and BC finetuning, pretrained models have a warm start, a faster convergence rate, and a relatively better asymptotic performance in a variety of downstream tasks. In most cases, pretraining CT from multi-task dataset (SMART) yields better results than pretraining with only in-task data (`CT-single`), although it is harder to accommodate multiple different tasks with the same model capacity, which suggests that SMART can extract common knowledge from diverse tasks.

Next, we show the **generalizability** of SMART. Figure 4 shows the performance of SMART pretrained on `Exploratory` dataset[3], compared to `Scratch` and `CT-single` on 5 unseen tasks. We can see that SMART is able to generalize to unseen tasks and even unseen domains, whose distributions have a larger discrepancy as compared to the pretraining dataset. Surprisingly, SMART achieves better performance than `CT-single` in most tasks, even though `CT-single` has already seen the downstream environments. This suggests that good generalization ability can be obtained from learning underlying information which might be shared among multiple tasks and domains,

---

[3]Models pretrained with `Random` dataset show similar results, as can be seen in Appendix B.1

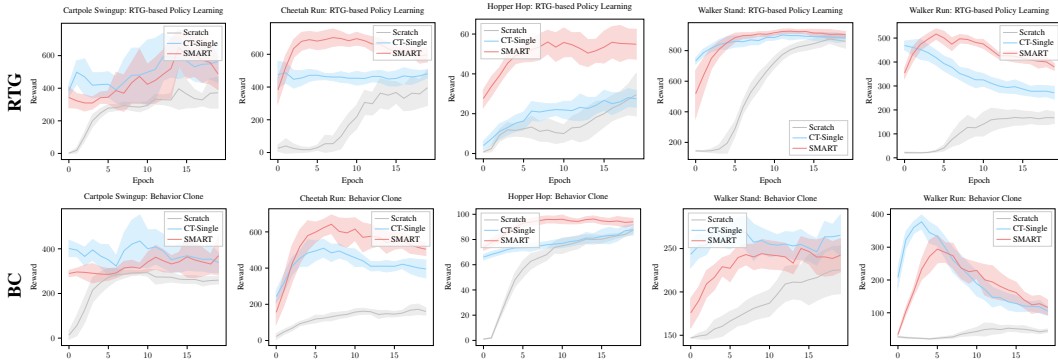

**Figure 3:** Downstream learning rewards of SMART (red) compared with pretraining CT with single-task data (blue) and training from scratch (gray). Results are averaged over 3 random seeds.

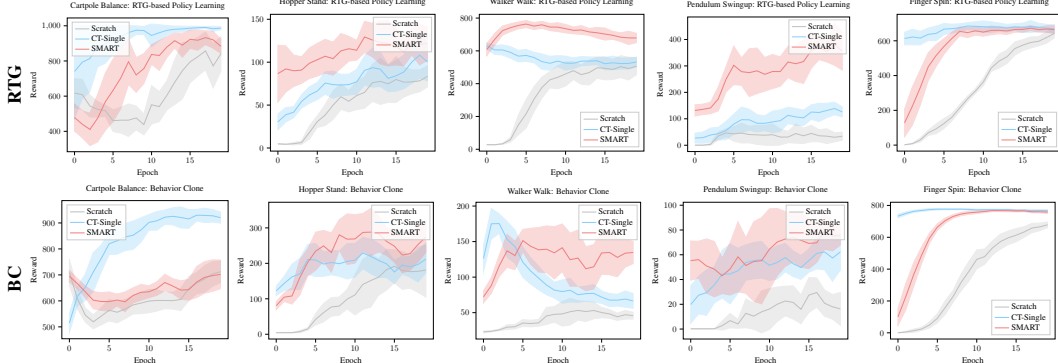

**Figure 4:** Downstream learning rewards **in unseen tasks and domains** of SMART (red) compared with pretraining CT with single-task data (blue) and training from scratch (gray). Results are averaged over 3 seeds.

spanning a diverse set of distributions. Appendix B.1 provides additional results in other challenging tasks which have a larger discrepancy with pretraining tasks, where SMART still performs well.

Next we evaluate the **resilience** of SMART by comparing with all aforementioned baselines, as visualized in Figure 5. We aggregate the results in all tasks by averaging the normalized reward (dividing raw scores by expert scores) in both RTG and BC settings. When using the `Exploratory` dataset for pretraining, SMART outperforms `ACL`, and is comparable to `DT` which has extra information of reward. When pretrained with the `Random` dataset, SMART is significantly better than `DT` and `ACL`, while `ACL` fails to outperform training from scratch. This result show that SMART is robust to low-quality data as compared to other baseline methods.

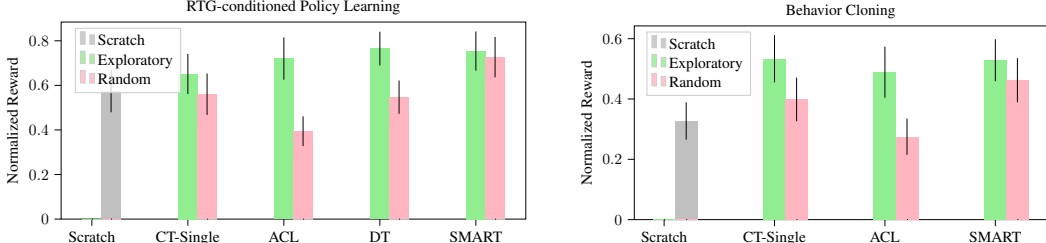

**Figure 5:** Downstream learning rewards (normalized by expert score) of all methods using `Exploratory` and `Random` dataset. The gap between each pair of green and red bars corresponds to the resilience of each method to pretraining data quality, and our SMART shows the best resilience among all baselines.

## 6.3 ABLATION AND DISCUSSION

**Ablation of Pretraining Objectives.** As we analyzed in Section 5.2, forward prediction and inverse prediction aim to learn the information of short-term control, while the random masked hindsight control (in short, Mask-Ctl) learns the long-term control information. To show the effectiveness of combining these two kinds of information, we conduct ablation study over these three terms. Figure 6 demonstrates their relative improvements wrt `Scratch` as defined in Appendix A.4. We

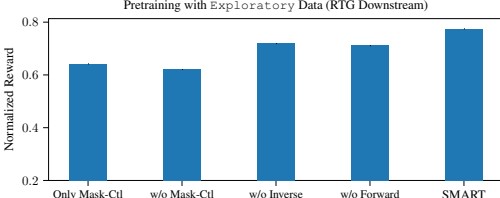
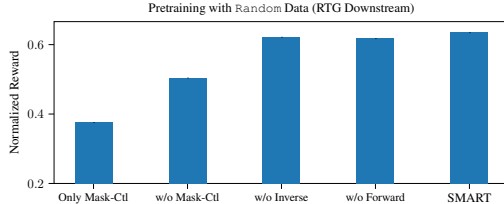

**Figure 6:** Ablation study on proposed pretraining objective. Rewards are averaged over tasks. Both long-term control information (Mask-Ctl) and short-term control information (Forward and Inverse) are important.

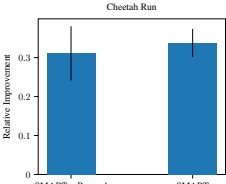
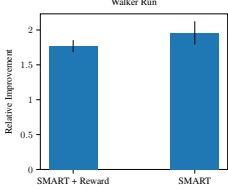
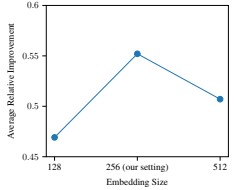
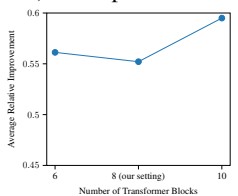

(a) Reward-based v.s. reward-free pretraining.  (b) Overall performance v.s. model capacity.

**Figure 7:** Ablation study of the effect of reward in pretraining and comparison of various model capacity.

can see that only learning long-term control information (Only Mask-Ctl) or only learning short-term control information (w/o Mask-Ctl) renders much lower performance than the original SMART that leverages both types of information. In addition, we find that both the performance slightly drops if either forward prediction or inverse prediction is removed, as the combination of these two terms leads to more stable and comprehensive representation for short-term control. This ablation study verifies the effectiveness of each component in the proposed pretraining objective. More results in other settings are provided in Appendix B.5.1.

**Reward-free v.s. Reward-based Pretraining.** As discussed in Section 5.1, including reward information in pretraining objectives is not necessarily helpful. We study the effects of rewards in pretraining by adding two reward-based objectives in the pretraining phase: immediate reward prediction and RTG-based action prediction. We evaluate this reward-based variant of SMART using exploratory dataset and RTG-conditioned downstream learning. Note that the RTG-based action prediction is used both in pretraining and finetuning of the reward-based variant, providing strong supervision for the pretrained model. However, we (surprisingly) observe that such supervision does not improve the downstream performance in many tasks, as shown in Figure 7a. A potential reason is that reward-based objectives are more fragile to distribution shifts, which also explains the non-ideal performance of DT pretrained from random data.

**Discussion on Model Capacity.** In large-scale training problems, performance usually benefits from larger model capacity (Kaplan et al., 2020). We investigate if this also applies to sequential decision making tasks by varying the embedding size (width) and the number of layers (depth) in CT. The aggregated results averaged over all tasks are show in Figure 7b. From the comparison, we can see that in general, increasing the model depth leads to a better performance. However, when embedding size gets too large, the performance further drops, as a large representation space might allow for irrelevant information. Per-task comparison is provided in Appendix B.2.

**Other Potential Variants of SMART.** Following the basic idea of SMART, there could be many other variants and extensions on the design of both architecture and pretraining objective (e.g., does an additional contrastive loss help?). We investigate several variants of SMART and discuss the results in Appendix B.5.2. In summary, adding more components to SMART could further improve its performance in certain scenarios, although such improvement may be limited and task-dependent. Exploring more extensions of SMART in various settings is an interesting future work.

## 7 CONCLUSION

This paper studies how to pretrain a versatile, generalizable and resilient representation model for multi-task sequential decision making. We propose a self-supervised and control-centric objective that encourages the transformer-based model to capture control-relevant representation. Empirical results in multiple domains and tasks demonstrate the effectiveness of the proposed method, as well as its robustness to distribution shift and low-quality data. Future work includes strengthen the attention mechanism on both spatial observation space and temporal state-observation interactions, as well as investigating its potential generalization in a wider range of application scenarios.

## 8 ETHICS STATEMENT

SMART is a method that learns pretrained representations for controls tasks involving sequential decision-making. As such, our models will reproduce patterns and biases found in its pretraining and finetuning datasets.

## 9 REPRODUCIBILITY STATEMENT

The codebase, pretrained models and datasets are provided at https://github.com/microsoft/smart.

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

# Appendix

## A  IMPLEMENTATION DETAILS

### A.1  IMPLEMENTATION OF PRETRAINING OBJECTIVES

**I. Forward Dynamics Prediction.**
For each observation-action pair $(o_t, a_t)$ in a control sequence, the forward prediction loss is constructed as follows. Let $g_\theta$ be the observation tokenizer being trained where $\theta$ denotes the parameterization. We maintain a momentum encoder $\bar{g}_{\bar{\theta}}$ whose parameters are updated by $\bar{\theta} = \tau(\bar{\theta}) + (1-\tau)\theta$. With the next observation $o_{t+1}$, we have $\hat{s}_{t+1} := \mathsf{SG}(\bar{g}_{\bar{\theta}}(o_{t+1}))$, where $\mathsf{SG}$ refers to stop gradient. Then the forward prediction loss is defined as:

$$L_{\text{fwd}} := \text{MSE}\left(f_{\text{fwd}}(\phi(o_t), \phi(a_t)), \hat{s}_{t+1}\right), \tag{1}$$

where $f_{\text{fwd}}$ is a linear prediction head.

**II. Inverse Dynamics Prediction.**
For each consecutive observation pair $(o_t, o_{t+1})$, inverse prediction tries to recover the action that leads $o_t$ to $o_{t+1}$, which gives the loss function

$$L_{\text{inv}} := \text{MSE}\left(f_{\text{inv}}(\phi(o_t), \phi(o_{t+1})), a_t\right), \tag{2}$$

where $f_{\text{inv}}$ is a linear prediction head.

**III. Random Masked Hindsight Control.**
In our implementation, we use a predefined mask-token $\mathbf{m} = [-1, \cdots, -1]$ to replace the original tokens. Let $\tilde{h}$ denote the masked control sequence, and $0 \leq s_1, s_2, \cdots, s_k \leq L$ be the selected indices for masked actions. Then the loss function can be defined as:

$$
\begin{aligned}
L_{\text{mask-inv}} &:= \sum\nolimits_{i=1}^{k} l(s_i), \\
\text{where} \quad l(s_i) &= \begin{cases} \text{MSE}\left(f_{\text{mask-inv}}(\phi_M(\mathbf{m}_{t+s_i}); \tilde{h}), a_{t+s_i}\right), & \text{if } s_i < L \\ 0 & \text{if } s_i = L \end{cases}
\end{aligned}
\tag{3}
$$

where $f_{\text{mask-inv}}$ is a linear prediction head, and $\phi_M$ is the transformer model without a causal mask. Note that we do not predict $a_{t+L}$ as it is infeasible to recover it without future observations.

*Schedule of Masking Size.* Theoretically, it is possible to recover a full action sequence for a given observation sequence, which implies that $k = L$ is a reasonable setup. But in environments with complex dynamics, directly recovering all actions is hard in the start of training. Hence, we adjust the difficulty of random masked hindsight control in a curriculum way, by gradually increasing the value of $k$ in the following schedule:

$$k = \max\left(1, \text{int}\left(L * \frac{\text{current\_epoch} + 1}{\text{total\_epochs}}\right)\right) \tag{4}$$

On the other hand, if we predict actions from all observations, it is possible that the model mainly relies on local dynamics, i.e., predict $a_t$ mainly based on $o_{t-1}, o_{t-2}, o_{t-3}$, conflicting with our desire of learning long-term dependence. Therefore, we also mask a subset of observations with $k'$. In the extreme case $k' = L$, the objective becomes similar to goal-conditioned modeling with $o_t$ being the start and $o_{t+L}$ is the goal. However, in a control environment, there usually exist multiple paths from $o_t$ to $o_{t+L}$, making the action prediction ambiguous. Schwarzer et al. (2021) uses a goal-conditioned objective which finds the shortest path. However, extra value fitting and planning are required, which may lead to higher cost and compounding errors. Therefore, we intentionally make $k'$ smaller than the context length (half of $L$), such that the model is able to predict the masked actions based on revealed subsequence of observations and actions. The schedule of $k'$ is as below.

$$k' = \max\left(1, \text{int}\left(\frac{L}{2} * \frac{\text{current\_epoch} + 1}{\text{total\_epochs}}\right)\right). \tag{5}$$

The overall pretraining objective is

$$\min_{\phi, f_{\text{fwd}}, f_{\text{inv}}, f_{\text{mask-inv}}} L_{\text{fwd}} + L_{\text{inv}} + L_{\text{mask-inv}}. \tag{6}$$

## A.2 ENVIRONMENT AND DATASET.

Table 2 lists all domains and tasks from DMC used in our experiments, and their relations are further depicted in Figure 8.

| Phase | Domain | Task | Expert Score by SAC |
|---|---|---|---|
| Pretraining & Finetuning | cartpole | swingup | 875 |
| | hopper | hop | 200 |
| | cheetah | run | 850 |
| | walker | stand | 980 |
| | walker | run | 700 |
| Finetuning only | cartpole | balance | 1000 |
| | hopper | stand | 900 |
| | walker | walk | 950 |
| | pendulum | swingup | 1000 |
| | finger | spin | 800 |

**Table 2:** A list of domains and tasks used in pretraining and finetuning. The first 5 tasks are used for pretraining. In the finetuning phase, we use the pretrained model to learn policies in all 10 tasks, including the last 5 tasks that are unseen during pretraining.

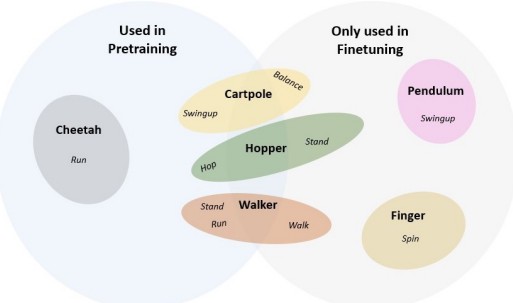

**Figure 8:** Graphical relations of all tasks involved.

To generate our datasets for pretraining and finetuning, we first train 5 agents (corresponding to different random seeds) for each task with ground truth vector states using SAC (Haarnoja et al., 2018) for 1M steps, and collect the full replay buffer with corresponding RGB images ($3 \times 84 \times 84$) rendered by the physical simulator. Then, we divide the replay buffers and create the following datasets of different qualities.

- `Random`: Randomly generated interaction trajectories. This dataset has 400K timesteps per task.
- `Exploratory`: The first 80K timesteps of each SAC learner, corresponding to the exploratory stage. For all 5 agents, this leads to a cumulative of 400K timesteps per task.

Note that different tasks have different difficulties for the SAC agent to converge. For example, in cartpole, the agent converges with much less samples than in walker. Therefore, we slightly adjust the proportion of data from different tasks in multi-task pretraining to avoid overfitting to simple tasks and underfitting to harder tasks. In pretraining, we use 40K timesteps from each cartpole behavior agent (200K by 5 agents), and 90K timesteps from each walker behavior agent (450K by 5 agents), and 80K for all other agents (400K by 5 agents).

Datasets for downstream learning:

- `Sampled Replay` (for RTG): We randomly sample 10% trajectories from the full replay buffer of 1 SAC agent, resulting in a dataset of size 100K per task, with diverse return distribution.
- `Expert` (for BC): We select 10% trajectories with the highest returns from the full replay buffer of 1 SAC agent, resulting in an expert dataset of size 100K per task.

## A.3 MODEL AND HYPERPARAMETERS.

Following the implementation of Decision Transformer (Chen et al., 2021), our transformer backbone is based on the minGPT implementation https://github.com/karpathy/minGPT with the default

AdamW optimizer (Loshchilov & Hutter, 2019). For both pretraining and finetuning, the learning rate is set to be $6 \times 10^{-4}$ and batch size 256. For learning rate, linear warmup and cosine decay are used. A context length 30 is used in all tasks for both training and execution. We tested different context lengths in preliminary experiments, and a shorter context length (5/10/20) does not work as well as 30 when training from scratch. We use 8 attention heads, 8 layer attention blocks, and an embedding size 256 in all experiments, for both our model and baselines. There are 10.8 M trainable parameters in the model. We test SMART with varying layer numbers and embedding sizes in Appendix B.2.

For the execution of learned RTG-conditioned policies, we set the expected RTG as the expert scores in Table 2. Tuning the RTG setting may further increase the results. But since our focus is to show the effectiveness of pretraining, we did not explore other possibilities.

All models are trained for 10 epochs in pretraining, and 20 epochs for each downstream task. The performance of the best checkpoint is reported, as detailed in the next section.

### A.4 EVALUATION METRICS.

For both BC and RTG downstream learning, we report the average cumulative reward of the learned policy by interacting with the environment for 50 episodes. In Figure 5, we calculate the *normalized reward* based on expert scores.

In Figure 6 and Figure 7a, we report the relative improvement of each ablated method calculated by the following formula.

$$\text{relative\_improvement} := \frac{\text{method\_reward} - \text{scratch\_reward}}{\text{scratch\_reward}}, \tag{7}$$

where the scratch_reward is the best reward of training from scratch using the same learning configurations.

## B ADDITIONAL EXPERIMENT RESULTS

### B.1 MORE RESULTS OF DOWNSTREAM LEARNING

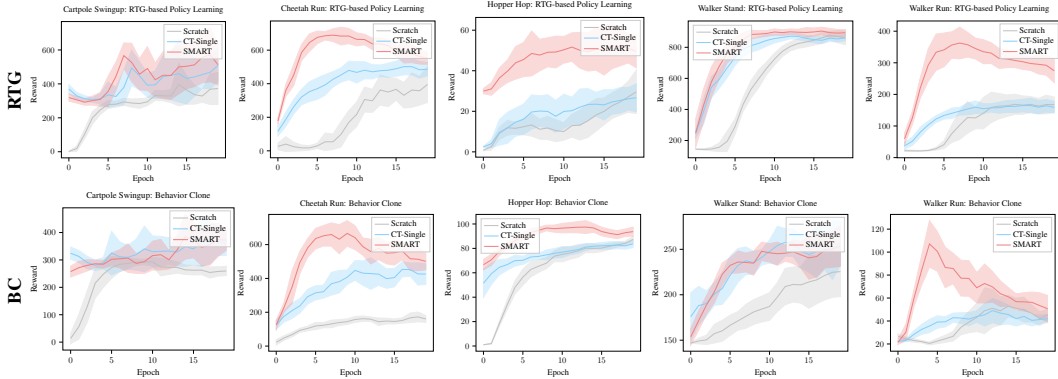

**Figure 9:** Downstream learning rewards of SMART (red) compared with pretraining CT with single-task data (blue) and training from scratch (gray), using the `Random` pretraining dataset. Results are averaged over 3 random seeds.

**Curves with the `Random` Dataset.** The results in Figure 3 and Figure 4 are generated with the `Exploratory` pretraining dataset. Now, we show the performance of models pretrained using the `Random` dataset in seen tasks and unseen tasks in Figure 9 and Figure 10, respectively.

Although the single-task pretrained model consistently outperforms training from scratch when pretrained with the `Exploratory` dataset, it sometime gets worst-than-scratch downstream performance when pretrained with the Random dataset. Therefore, it is challenging to overcome the distribution shift problem. In contrast, SMART still achieves much better performance than training from

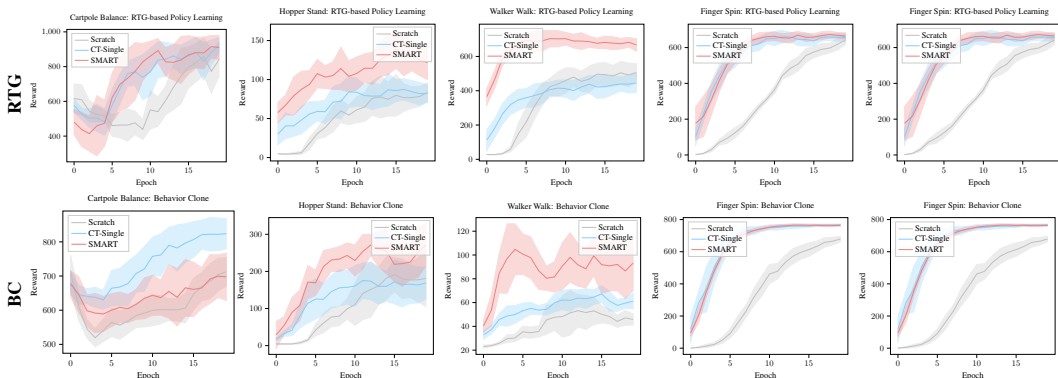

**Figure 10:** Downstream learning rewards in **unseen tasks** and domains of SMART (red) compared with pretraining CT with single-task data (blue) and training from scratch (gray), using the `Random` pretraining dataset. Results are averaged over 3 random seeds.

scratch in all tested tasks, which verifies the resilience of SMART due to multi-task self-supervised pretraining.

**Generalizability of SMART: Tasks That Have Larger Discrepancy with Pretraining Tasks.** In addition to the generalizability test we show in Section 6.2, we evaluate the performance of the pretrained SMART model in other more challenging domains/tasks from DMC, that are significantly different from the pretraining tasks. These additional domain-tasks are: ball-in-cup-catch, finger-turn-hard, fish-swim, swimmer-swimmer6 and swimmer-swimmer15. Note that these agents have significantly different appearance and moving patterns compared to pretraining tasks, as visualized in Figure 11.

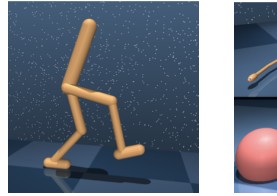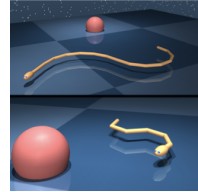

**Figure 11:** Discrepancy between pretraining domains and selected downstream domains: **(left)** Walker domain. **(right)** Swimmer domain (6 and 15 links)

The results are shown in Figure 12 and Figure 13, where we can see that the pretrained model can still work in most cases, even under such a large task discrepancy. Note that here `CT-Single` is pretrained with data from exactly the downstream task, where SMART has never seen a sample from the downstream tasks and is pretrained on significantly different domains. Therefore, it is unsurprising that `CT-Single` is generally better than SMART in this setting. However, it is interesting to see that SMART is comparable with or even better than `CT-Single` in some tasks, suggesting the strong generalizability of SMART.

On the other hand, one can imagine that it is unavoidable that the performance of a pretrained model will decrease as the discrepancy between pretraining tasks and downstream tasks increases. Therefore, we stress the importance of using diverse multi-task data for pretraining in practice.

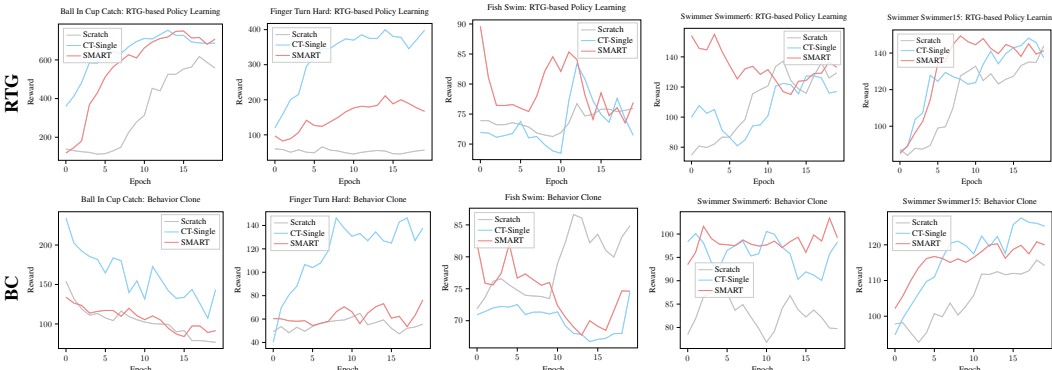

**Figure 12:** Downstream learning rewards of SMART (red) in challenging tasks that have larger discrepancy with pretraining tasks, using the `Exploratory` pretraining dataset. Results are from 1 random seed.

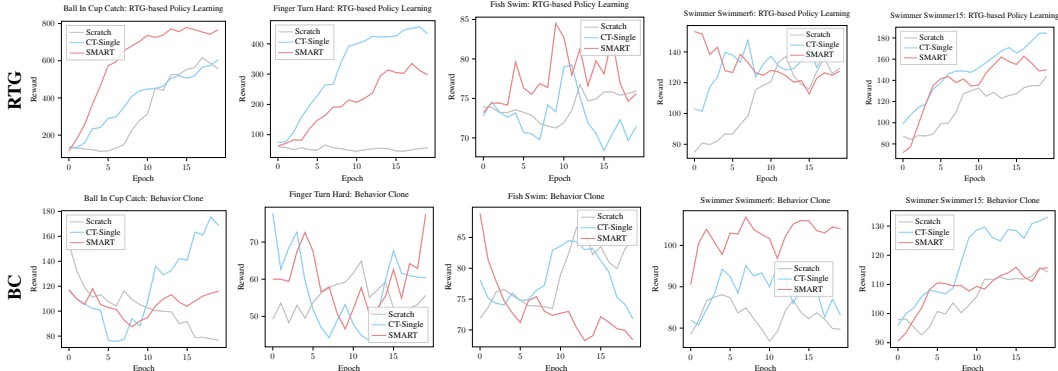

**Figure 13:** Downstream learning rewards of SMART (red) in challenging tasks that have larger discrepancy with pretraining tasks, using the `Random` pretraining dataset. Results are from 1 random seed.

## B.2 MODEL CAPACITY TEST

Figure 7b shows the results of varying model capacities averaged over tasks. To demonstrate the influence of model capacity on different tasks, we provide the per-task comparison in Figure 14.

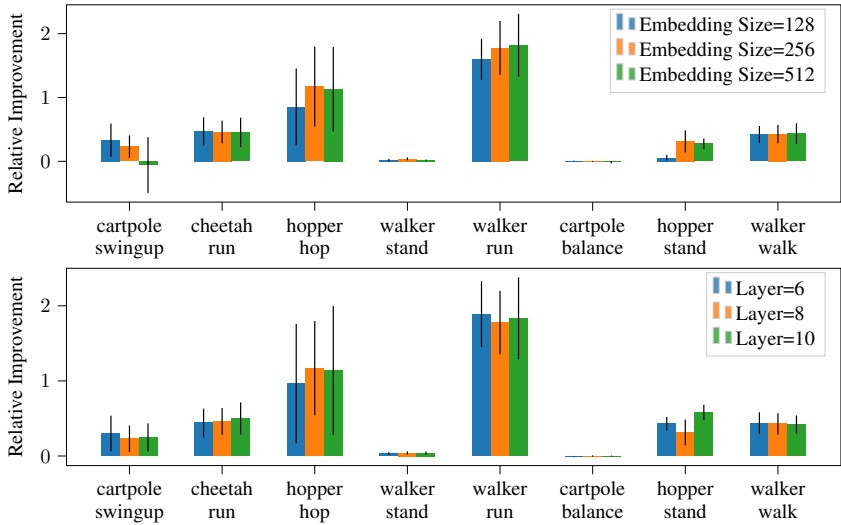

**Figure 14:** Comparison of varying model capacities (embedding size and layer number) in different tasks in terms of relative improvement wrt training from scratch.

**Further Decreasing Model Capacity.** In Figure 7b, we can see that a 6-layer transformer backbone is slightly better than the 8-layer transformer. It may suggest that the current network we use is overly large for the tasks. To see whether it is the case, we further decrease the number of transformer layers to 4 and 2. The results are depicted in Figure 15, where we can see that a 4-layer or a 2-layer transformer is clearly worse than the 6-layer and 8-layer transformers. Therefore, the current architecture is not too heavy for the DMC tasks. The exact numbers in Figure 15 are different from the corresponding ones in Figure 7b, because Figure 7b aggregates results from 3 random seeds but Figure 15 is from 1 random seed.

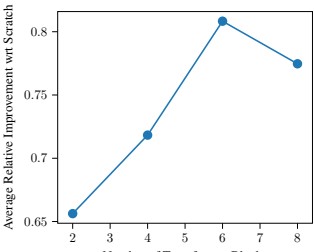

**Figure 15:** More model capacity tests. Results are from one random seed and are averaged over tasks.

## B.3 COMPARISON WITH PRETRAINED RESNET MODELS

Our empirical evaluation is done for multiple transformer-based pretraining approaches. In parallel to them, there are some existing pretraining paradigms that use large-capacity models as ResNet instead of transformers as the backbones. Although it is hard to directly compare the performance of totally different model architectures, we still provide the results of ResNet pretraining, to better posit this work in literature and verify the significance of our results.

**Baselines and Implementation Details.** We use the following two state-of-the-art pretraining approaches.

- CPC (Oord et al., 2018) is a self-supervised representation learning approach with contrastive predictive coding. It has been demonstrated success in many vision applications. When leveraging it in sequential decision making, state representations can be pretrained by decoupled from policy learning.
- ATC (Stooke et al., 2021) is another contrastive learning approach target on decision making tasks. It also decouples representation learning from policy learning. Different from CPC which performs InfoNCE (Oord et al., 2018) loss in a predictive manner, ATC propose an Augmented Temporal Contrast to directly compute InfoNCE loss among temporally augmented clips.

For both CPC and ATC, we use 3D-ResNet18 as the encoder backbone. During pretraining, a 2-layer 2D-ConvNet is used as the prediction head for both CPC and ATC. For CPC, a single-layer ConvGRU is used as the aggregation network. Note that, during finetuning, both prediction heads and aggregation network are dropped. Only the pretrained encoder (3D-ResNet18) is used to produce the pretrained representations. During finetuning, we simply attach a single linear layer as the action prediction head on top of the pretrained encoder, which is under the same setting with ours.

For a fair comparison, we pretrain and finetune both CPC and ATC with a context length of 30 such that all comparing models are seeing the same time horizon. Follow the widely used training protocol in decision making tasks, we also leverage a frame stacking with stack size as 3 when training both of them. We keep the other hyperparameters the same with our default setting.

**Results.** We compare our SMART with CPC and ATC in RTG and BC downstream tasks, with Exploratory and Random dataset. The results are shown in Figure 16 and Figure 17, respectively. Although training a transformer and training a ResNet model from scratch usually produces different rewards in the same task, the results show that pretrained models are able to their corresponding train-from-scratch baselines. When using RTG downstream learning, we can see that SMART outperforms CPC and ATC in all downstream tasks. When using BC as downstream learning method, the transformer backbone fails to get a high score and so does SMART pretrained models. But in remaining tasks, SMART is still significantly better than CPC and ATC. This suggests the advantages of our pretraining framework SMART and the proposed model Control Transformer.

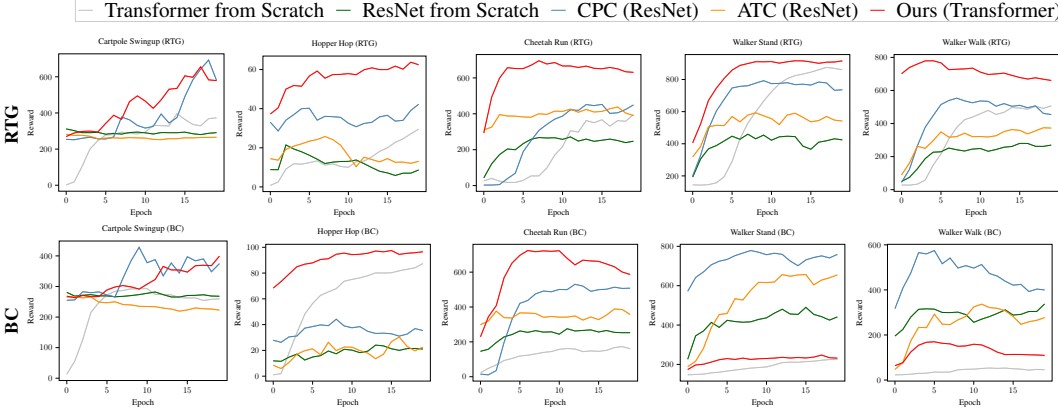

**Figure 16:** Downstream learning rewards of SMART (red) compared with CPC (darkblue) and ATC (darkorange), using the Exploratory pretraining dataset. CPC and ATC are based on ResNet, whose performance of training from scratch is shown in darkgreen in comparison to training transformer from scratch (grey).

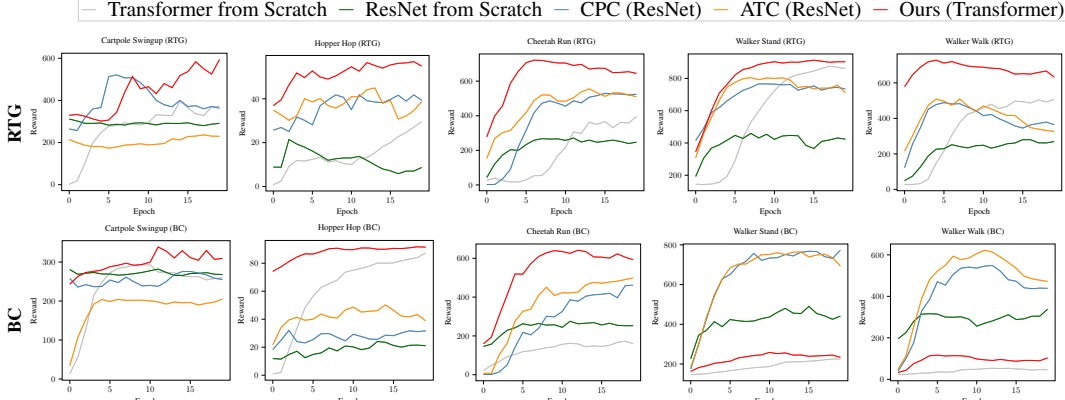

**Figure 17:** Comparison with ResNet-based pretrained models using the `Random` pretraining dataset.

## B.4 PRETRAIN CT FOR ONLINE FINETUNING

Similar to most transformer-based decision making models, we consider policy learning in the form of IL and offline RL. For online RL, special care is needed due to the environment uncertainty and the need of exploration. A recent work by Zheng et al. (2022) proposes online decision transformer (ODT), which first pretrains the model with offline trajectories and then finetune the model in an online manner. Zheng et al. (2022) demonstrate the effectiveness of ODT in a series of MuJoCo tasks with groundtruth state being observations. It is shown that the offline pretraining phase is crucial for online fintuning with transformer-based models. However, the pretraining phase of ODT is DT with reward supervision. With the self-supervised control-centric pretraining objective ($L_{\text{fwd}}$, $L_{\text{inv}}$, $L_{\text{mask-inv}}$) proposed in this work, we would like to ask the following questions. (1) Can we replace the supervised DT pretraining objective with our self-supervised pretraining objective that does not require reward supervision? (2) Can we improve ODT by combining our objectives with it?

We follow the open-sourced implementation of ODT and evaluate our proposed objective using the default model and hyperparameter settings of ODT. We first replace the DT pretraining loss with our self-supervised losses. The results shown in Figure 18 suggests that even without any reward information, pretraining ODT with our self-supervised losses can achieve comparable performance with pretraining with reward supervisions. Although our pretrained model takes more steps to warm up due to the lack of supervised pretraining, it quickly converges to similar results as ODT in online finetuning. Therefore, in practical applications where only reward-free pretraining trajectories are available, DT pretraining is infeasible while our pretraining can be used without sacrificing the finetuning performance.

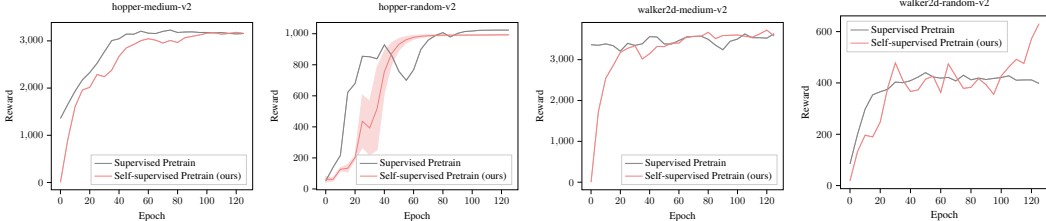

**Figure 18:** Comparison of our self-supervised pretraining objective and the supervised pretraining objective of ODT in hopper and walker2d environments with medium and random pretraining dataset.

To answer the second question, we combine our pretraining objective with ODT. There are two potential ways of such combination: (1) only combine in pretraining, and (2) add our self-supervised losses during both pretraining and finetuning (the losses serve as auxiliary tasks in online finetuning). We evaluate both versions in our experiments, as shown in Figure 19. The results suggest that including our pretraining objective can improve the performance of ODT, especially when the data quality is low ({task}-random dataset is used). However, when our objective is only used during pretraining, it suffers from high variance in downstream learning. To address this issue, we find that

adding our objective as auxiliary loss in finetuning can effectively improve the result and decrease the variance.

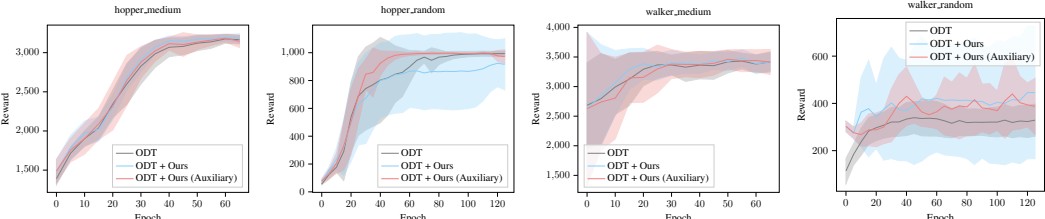

**Figure 19:** Combining our pretraining objectives with ODT produces better results, especially when the pretraining dataset has low quality. Results are averaged over 5 random seeds.

It is also interesting to see that the online learning performance of ODT depends on the quality of pretraining dataset. When the random dataset is used, ODT converges to a suboptimal policy in finetuning, although it could have explored better solutions. Adding our self-supervised pretraining objective improves the performance, but it still cannot match the performance of models pretrained in a higher-quality dataset. We hypothesize that this is due to a very large distribution shift between the random dataset and the ideal dataset for online learning. That is, the model pretrained with random data overfits to the random behavior and fails to further explore and exploit. How to make a transformer-based model adapt to an online environment with low-quality pretraining data is still an open problem that we aim to study in our future work.

## B.5 ADDITIONAL RESULTS OF ABLATION STUDY

### B.5.1 ABLATION OF PRETRAINING OBJECTIVE

In addition to Figure 6 which uses RTG as the downstream learning objective, we also provide the ablation results of BC downstream learning in Figure 20. Similar to the results of RTG, removing any term from the control-centric pretraining objective usually causes a performance drop, which verifies the effectiveness of our objective design. One exception is that removing the inverse prediction gives a better result for BC with exploratory pretraining data. Combined with results in other cases, we can find that inverse prediction is not as helpful as forward prediction and random masked hindsight control on average. But in most cases, especially when pretraining data is low-quality (`Random`), including the inverse prediction still improves the performance.

**Implementation Details.** For the ablation study, we pretrain the model with the same hyperparameter settings on the same datasets using different versions of (ablated) pretraining objectives. The results are averaged over 8 tasks: cartpole-swingup, cartpole-balance, hopper-hop, hopper-stand, cheetah-run, walker-stand walker-run, and walker-walk, spanning both seen and unseen tasks. Due to the large number of experiments (4 variants $\times$ 2 learning scenarios $\times$ 2 dataset selections $\times$ 8 downstream tasks = 128 experiments), the results in Figure 6 and Figure 20 are from one random seed. Although randomness exists, we believe that the conclusion is meaningful as the results are averaged over multiple tasks and multiple learning scenarios.

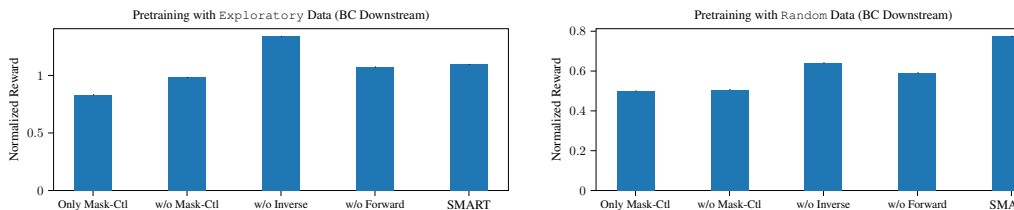

**Figure 20:** Ablation study of our pretraining objective with BC (behavior cloning) being the downstream learning algorithm. Both long-term control information (Mask-Ctl) and short-term control information (Forward and Inverse) are important.

### B.5.2 VARIANTS OF PRETRAINING OBJECTIVE

We also investigate some other possible variants of the pretraining objective.
**(1) Multi-step Inverse** (inspired by Lamb et al. (2022)): masking all states/actions but $s_t$ and $s_{t+L}$

and predicting $a_t$.

**(2) Max Fixed Mask**: using a fixed mask size $k = L$ and $k' = L/2$ instead of gradually increasing $k$ and $k'$.

**(3) + Masked State Prediction**: in addition to the original objective, this variant also predicts the masked state token in the third term. (The state token target is generated from the momentum encoder, as in forward prediction.)

**(4) + Contrastive Loss**: in addition to the original objective, this variant also employs an additional contrastive loss, which takes the state/action token and their representations for the same timestep as positive pairs, while regarding representations from other timesteps as negative samples (inspired by Banino et al. (2022)).

The implementation and comparison of these variants follow the same setup with the ablation study in Appendix B.5.1. Note that (3) and (4) are both adding new losses to the original method, which renders higher computational cost.

The results of comparison with these variants are shown in Figure 21, with RTG being the downstream learning objective. The results show that Multi-step Inverse and Max Fixed Mask are in general worse than SMART, suggesting the effectiveness of our design. Interestingly, optimizing an extra state prediction loss or a contrastive loss gives better performance when the pretraining data is `Random`, while hurts the performance when the pretraining data is `Exploratory`. We hypothesize that this is because such two additional losses emphasize more on capturing visual information from pixel observations, which is more important for learning and transferring representation from the `Random` dataset. In other words, a better visual representation is more crucial under a larger distribution shift (on behavior policies) that happened when a `Random` dataset is used for pretraining. On the other hand, when the pretraining data is of higher quality, adding these two components downgrades the original SMART. A potential reason is that with a smaller distribution shift, the control-centric objective can already capture sufficient information from data, while adding new losses may make the optimization harder and less stable.

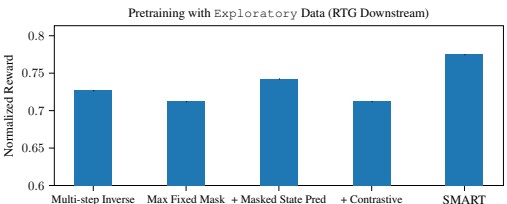 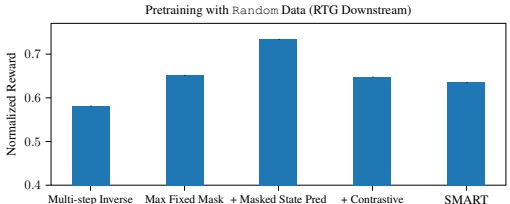

**Figure 21:** Comparison between SMART and its variants. Adding masked state prediction and contrastive loss to SMART could increase the performance, but the improvement is not across all scenarios, with a cost of more computations.

In summary, although none of the tested variants can outperform SMART in all scenarios, some of them can render better results in certain cases by adding extra pretraining losses to the original objective, at a cost of more computations and possibly more challenging optimization. Therefore, whether to use these variants in practice depends on the specific use case and computation resources. A deeper and more thorough understanding of the relationships among these pretraining losses is desired for better application of pretraining methods, which would be our future work. We emphasize that multi-task pretraining for control tasks is a relatively new area. This paper takes one step further towards general large-scale pretraining models for sequential decision making. Exploring more possibilities of SMART would be an interesting and important future direction to further improve its performance in a variety of real-world tasks.

