# OpenReview forum: "SMART: Self-supervised Multi-task pretrAining with contRol Transformers"
_ICLR.cc/2023/Conference — ICLR 2023 notable top 25%_

### Official Review · Reviewer_Jx8K · 2022-10-24

**Confidence:** 4
**Correctness:** 4
**Technical Novelty And Significance:** 3
**Empirical Novelty And Significance:** 3
**Recommendation:** 8

**Clarity, Quality, Novelty And Reproducibility:**

Quality
- Paper proposes a simple and effective method for unsupervised pretraining using Transformer.
- The idea is novel and simple to implement. The idea is to pretrain a Transformer in unlabelled datasets by predicting the masked tokens.
- The experimental results show the effectiveness, SMART outperforms baselines in multiple settings. The evaluations are extensive and convincing.

Clarity
- Mostly very clear. The key concepts are well explained and main claims are well supported.
- How did the authors decide to choose which tasks and domains for pretraining and which for finetuning? I appreciate the authors present the Figure 8 to illustrate the relationship between pretraining and finetuning tasks but it’s unclear the reason behind the decision.


**Strength And Weaknesses:**

Strength
- This work proposes a simple and effective method, for unsupervised pretraining and adaptation to downstream imitation learning and RL tasks.
- The experiments are extensive and show the strength of SMART, and I appreciate the authors conduct pretraining experiments on random and exploratory datasets since otherwise simple behavior cloning would be sufficient.

Weakness
- How does SMART perform in goal conditioned tasks and model-based tasks? Showing the results could help have a better understanding of what SMART captures from pretraining, especially since it’s trained with forward, inverse dynamic prediction and random hindsight prediction.
- The Figure 11 scalability experiment is interesting because it shows after a certain amount of model capacity, there is no further relative improvement. I wonder if the authors tried smaller model sizes since the current smallest model is arguably quite large for simpler tasks.




**Summary Of The Paper:**

The authors consider the problem of unsupervised pretraining representation for  high-dimensional sequential control in various downstream tasks. The authors propose a framework as a pretraining then finetuning pipeline for sequential decision making, it consists of a Control Transformer (CT) which is coupled with a novel control-centric pretraining objective in a self-supervised manner. The authors show that the proposed method, SMART, significantly improves the learning efficiency in both seen and unseen downstream tasks and domains.

**Summary Of The Review:**

This paper tackles the problem of unsupervised pretraining for sequential decision making. The proposed algorithm is simple and effective, with extensive experiments to show its effectiveness. There are some minor limitations of this paper that I would like the authors to address.

---

> ### Author Response · Authors · 2022-11-18
> **Part II: Explanation on Task Selection**
>
> > **Q3:** How did the authors decide to choose which tasks and domains for pretraining and which for finetuning? I appreciate the authors present the Figure 8 to illustrate the relationship between pretraining and finetuning tasks but it’s unclear the reason behind the decision.
>
> **A3:**
> It is a nice point. We would like to first emphasize that all 10 tasks in Figure 8 are used in downstream finetuning. But some of them are used in pretraining while some are held out for downstream learning only. Therefore, we just need to select what tasks to use in pretraining, as explained below.
>
> Our main consideration is to **create convincing and rich testing scenarios** to evaluate SMART from different angles. Finally, we select tasks such that there are **3 types of downstream tasks with different levels of distribution shifts** between pretraining and finetuning:
>
> (1) The downstream task has been seen in pretraining, but with a different type of dataset (cheetah-run, cartpole-swingup, hopper-hop, walker-stand, walker-run).
>
> (2) The downstream task has not been seen in pretraining, but the corresponding domain has been seen in the form of different tasks (cartpole-balance, hopper-stand, walker-walk).
>
> (3) The downstream task has not been seen in pretraining, and the corresponding domain has not been seen, either (pendulum-swingup, finger-spin).
>
> Therefore, this design allows us to evaluate the versatility and generalizability of SMART with different levels of difficulties. The results suggest that SMART performs well in all of them.
>
> Another important standard for pretraining task selection is to include **diverse tasks**. This is common and intuitive, because one cannot expect a model that is pretrained on only one task to generalize to all other different tasks. Therefore, we select 5 pretraining tasks across 4 domains, with different task difficulties.
>
> With the above motivation, we use the current pretraining-task selection. But this is definitely not the only choice. **In practice**, we would recommend users to include diverse tasks and data in pretraining to obtain the best results. But note that more pretraining tasks would desire larger model capacity and more computation resources. So one can balance between diversity and efficiency in their real-world applications.
>
> ---
>
> Thank you again for spending time reviewing our paper and providing constructive suggestions. We hope our above answers have addressed all the questions. Please let me know if there are any other questions.

---

> ### Author Response · Authors · 2022-11-18
> **Part I: Discussion on Potential Extensions; Additional Results of Model Capacity**
>
> We greatly appreciate the valuable feedback of Reviewer Jx8K. We have added more experiments and discussions as suggested. Below we address the questions raised by Reviewer Jx8K.
>
> ---
>
> > **Q1:** How does SMART perform in goal conditioned tasks and model-based tasks? Showing the results could help have a better understanding of what SMART captures from pretraining, especially since it’s trained with forward, inverse dynamic prediction and random hindsight prediction.
>
> **A1:**
> It is a great suggestion to apply SMART to goal-conditioned tasks and model-based RL. It is reasonable to believe that the proposed control transformer architecture can help in these settings, given its ability to model both short-term control and long-term control information.
>
> For **goal-conditioned tasks**, it is particularly important for the model to capture long-term control information (i.e., which current action can finally lead us to the goal after many steps). Since the "random masked hindsight control" term of our objective is designed to help with such long-term control, we believe that SMART can be suitable for goal-conditioned problems. On the other hand, the random masked hindsight control can also be regarded as an implicit goal-conditioned planning process, and thus it has the potential to improve goal-conditioned policy learning (with the unmasked intermediate tokens as hints). We are trying to incorporate SMART into existing goal-conditioned testbeds. However, since it is a new setting with different environments/datasets, it is hard to obtain sufficient results within the limited rebuttal period. We would like to keep running it and put the results in the final version of the paper, or in our future work.
>
> The common practice of **model-based RL** is to learn a world model and train policies with samples generated by the model. With the ability of modeling transition relations within a long time window, SMART can be a good candidate for the world model. It would be an exciting future direction, but out of the scope of this paper which aims to pretrain a representation for downstream adaptation. We would like to explore the combination with model-based learning in our future work.
>
> ---
>
> > **Q2:** The Figure 11 scalability experiment is interesting because it shows after a certain amount of model capacity, there is no further relative improvement. I wonder if the authors tried smaller model sizes since the current smallest model is arguably quite large for simpler tasks.
>
> **A2:**
> Thank you for the insightful suggestion! We conducted **additional experiments** and put the results in Appendix B.2 of our revised paper. The new result in Figure 15 shows that with smaller model sizes (4-layer or 2-layer transformer backbone), the performance significantly drops. In comparison, we conclude that a 6-layer or 8-layer transformer has suitable model capacity in the selected tasks.
>
> Please note that SMART learns multiple tasks during pretraining, and can adapt to many different tasks in downstream learning. Even if some tasks are relatively simple, handling multiple tasks with a single model still requires a relatively large model capacity.
>
> For the representation embedding size, Figure 7(b) already shows that 256 is a suitable choice, and decreasing it to 128 causes an insufficient information bottleneck. Therefore, we did not further test smaller embedding sizes.

---

> ### Comment · Reviewer_Jx8K · 2022-12-11
> **Thanks for the response**
>
> I appreciate the authors' response which nicely address my questions.

---

### Official Review · Reviewer_CXdh · 2022-10-24

**Confidence:** 3
**Correctness:** 3
**Technical Novelty And Significance:** 3
**Empirical Novelty And Significance:** 3
**Recommendation:** 8

**Clarity, Quality, Novelty And Reproducibility:**

The clarity and quality of this work are good. The technical novelty is okay though a bit limited if considering the high-level idea similarity to those mask-based transformer approaches and the related control- or decision-based methods. As the code was uploaded, there should be no issue with reproducibility.

**Details Of Ethics Concerns:**

The authors pointed out the potential biases within the datasets in their ethics statement (Sec. 8). But this seems to be a common issue for most learning-based approaches, and no concern for an ethics review.

**Strength And Weaknesses:**

**Strengths**

\+ The paper well summarised the challenges unique to sequential decision-making scenarios and accordingly proposed a solution to address the issues.

\+ The idea of the proposed control transformer (CT) is interesting and well-motivated, especially the purposefully designed exclusion of the reward signal for pre-training.

\+ The experiments were well designed to validate the proposed method from several different perspectives in response to the questions mentioned at the beginning of Sec. 6.

\+ The proposed method was shown to have better resilience when facing pretraining data with different qualities, as shown in Fig. 5.

\+ The paper is well-written and easy to follow.


**Weaknesses**

\- The novelty of the proposed random masked hindsight control is a bit limited. Although the authors claimed the difference to BERT-like models, the general idea of mask-and-predict is the same and the only difference is the items to mask. This idea is also similar to those visual pre-training models like ViT etc.

\- It is unclear why only masking the actions while leaving out the observations. The authors claimed the reason to be "force the model to learn global temporal relations", but there is no clear evidence to support this.
What if including the observations as well, or partially? or what if similar to the random scheme as proposed, randomly predict the observations as well?

\- From the description, the overall objective used all three mentioned terms as mentioned in Sec. 5.2 / Fig. 2. The description also seems to suggest the effectiveness of the 3rd term of random masked hindsight control, but it is unclear how would the model performs if only using this term.

\- For the experiment of ‘versatility’ as shown in Fig. 3, there are some cases where the CT-single performs on par with or even better than the proposed SMART, i.e. single-task performs better. It would be better if some explanation or insights could have been provided.

**Summary Of The Paper:**

This paper presented a new self-supervised pre-training method for sequential decision-making tasks. Specifically, a new general pre-training-finetuning pipeline named SMART (Self-supervised multi-task pertaining with control transformer) was proposed with a Control Transformer (CT) and a control-centric pre-training objective for the model supervision. Experiments in DeepMind Control Suite show the effectiveness of the proposed method over several tasks. The main contributions of this paper are the proposed new self-supervised pre-training pipeline with the corresponding objective and also the carefully designed experimental analysis.

**Summary Of The Review:**

The studied problem is clearly stated with convincing motivations, the paper is well-written and the idea is mostly clearly presented. The experiments also well validate the claims with shown effectiveness of the proposed method over other solutions. Although there are a few concerns as mentioned in the Weaknesses section above, they could be addressed in the rebuttal phase, and overall this paper would be a good contribution to the community. As a result, I would like to recommend 'accept' for this paper.

-----------------After rebuttal---------------------

Thanks to the authors' responses, which well addressed my concerns. The additional included experiments and the clarification also make the paper stronger.

---

> ### Author Response · Authors · 2022-11-18
> **Part III: More Ablation Studies of SMART and Discussions on CT-single**
>
> > **Q3:** From the description, the overall objective used all three mentioned terms as mentioned in Sec. 5.2 / Fig. 2. The description also seems to suggest the effectiveness of the 3rd term of random masked hindsight control, but it is unclear how would the model performs if only using this term.
>
> **A3:**
> Thank you for this insightful point. We **added the suggested experiments** of only using random masked hindsight control. The results are shown in Figure 6 (Section 6.3) and Figure 20 (Appendix B.5.1), where the **"Only Mask-Ctl"** stands for the variant of only using the 3rd term. We can see that only using this term is worse than the original SMART. This suggests that both short-term control information (forward and inverse) and long-term control information (random masked hindsight control) are important to representation learning, which verifies our design of the control-centric pretraining objective.
>
> **Why is it crucial to learn both short-term and long-term information?** We note that learning only short-term information may lead to representation collapse as suggested by prior works[3,4]. On the other side, the long-term information is more policy-dependent, as it models sub-trajectories generated by some behavior policies. This is supported by the fact that the "Only Mask-Ctl" variant fails more significantly when the pretraining dataset is *Random* rather than *Exploratory*. In this case, including the forward and inverse prediction is critical, as they focus on the one-step transition dynamics $P(s^\prime|s,a)$, which is independent of the behavior policy and thus more resilient to low-quality data. On the other hand, it is intuitive that both local/finer temporal information and global/coarser temporal information should be taken into consideration when making control decisions. This is why we design the pretraining objective to include both types of information.
>
>
> ---
>
> > **Q4:** For the experiment of ‘versatility’ as shown in Fig. 3, there are some cases where the CT-single performs on par with or even better than the proposed SMART, i.e. single-task performs better. It would be better if some explanation or insights could have been provided.
>
> **A4:**
> It is a good point. Please note that CT-single and SMART have the **same model capacity and the same information bottleneck** (representation embedding size), but CT-single is performing a simpler job. CT-single is pretrained with data that is exactly from the downstream task, so the gap between pretraining and downstream learning is smaller. In contrast, SMART is pretrained with data from multiple tasks, and it has to accommodate tasks with diverse visual features and dynamics. Just as how a specialist usually outperforms a generalist, it is reasonable that CT-single can do better than SMART on a specific task in downstream learning.
>
> Based on the above insight, it is actually surprising to see that our SMART can outperform CT-single, even in tasks that SMART has not seen during pretraining (experiments for 'generalizability'). This phenomenon implies that SMART can learn common knowledge of the underlying task distribution and transfer the knowledge from task to task. It justifies the importance and feasibility of multi-task pretraining for reinforcement learning.
>
> We have added the explanation to our revised paper as suggested.
>
>
> ---
>
>
>
>
> Thank you again for your valuable feedback. We hope the above explanations and additional empirical results have addressed your questions. We are happy to discuss further if there are any other questions.
>
>
> ---
>
> Refs:
>
> [3] Rakelly, Kate, et al. "Which Mutual-Information Representation Learning Objectives are Sufficient for Control?." ICLR 2021.
>
> [4] Lamb, Alex, et al. Guaranteed discovery of controllable latent states with multi-step inverse models. arXiv preprint arXiv:2207.08229, 2022.

---

> > ### Comment · Reviewer_CXdh · 2022-12-12
> > **Thanks for the rebuttal**
> >
> > Thanks to the authors' responses, which well addressed my concerns. The additional included experiments and the clarification also make the paper stronger.

---

> ### Author Response · Authors · 2022-11-18
> **Part II: Predicting Masked Observations: Explanations and Experiments**
>
> > **Q2:** It is unclear why only masking the actions while leaving out the observations. The authors claimed the reason to be "force the model to learn global temporal relations", but there is no clear evidence to support this. What if including the observations as well, or partially? or what if similar to the random scheme as proposed, randomly predict the observations as well?
>
> **A2:**
> This is a great point. **We first clarify that we mask both actions and observations, while only predicting masked actions**. We explain the rationale and discuss the possible variants with empirical evidence from several aspects below.
>
> **1. Why masking observations when predicting actions?**
>
> In our preliminary experiments, we only randomly masked actions and left observations in the input, which also worked well. But a problem is that the model may predict $a_t$ only based on the closest observations, while not necessarily looking at transitions further away. Motivated by the idea of learning global temporal relations, we further added random observation masks. In this case, since $o_t$ and $o_{t+1}$ may be masked, the model has to look at remoter timesteps to infer $a_t$. We found that this implementation in general renders better results.
>
> **2. Why not also predicting observations?**
>
> First, observations are usually high-dimensional (e.g., images in DMC tasks), and contain irrelevant information (e.g., background texture), so reconstructing the raw observations can be expensive and result in redundant representation. In contrast, actions are relatively more informative. Note that the ultimate goal of downstream learning is to learn a policy to output actions. Therefore, it is less necessary to predict observations.
>
> Second, it is feasible to predict observation tokens/embeddings. However, since the observation tokenizer is jointly trained with the transformer model, predicting observation embeddings that are consistently changing could be hard and unstable. On the contrary, actions in the trajectories are available as groundtruth, so that action prediction is easier to learn.
>
> Due to the above reasons, we did not include observation prediction as part of our pretraining objective, and let the objective focus more on "how to control". However, it is still feasible, sometimes beneficial, to include observation prediction, as we will discuss in the next point.
>
> **3. What if we also predict observations?**
>
> Although we pointed out some challenges of observation prediction in bullet 2 above, they can be remedied in practice. For example, one can adopt a momentum encoder to generate more stable observation embeddings and the prediction targets, as what we use in forward prediction. Based on this trick, we implement a variant of SMART which also predicts masked observation embeddings in the 3rd term. The results are shown in Appendix B.5.2. The results show that this variant is sometimes better than SMART, while sometimes worse than SMART. With RTG downstream learning, this variant is better when *Random* pretraining data, but is worse when *Exploratory* data is used in pretraining. Detailed discussion and reasoning can be found in Appendix B.5.2.
>
> **In summary**, our empirical results show that including observation prediction may help or hurt, depending on the learning scenarios. We do not include it in our proposed objective because it does not consistently improve the performance, and requires extra computations. In practice, one can still consider predicting masked observations as an extra loss, especially when observations are low-dimensional. More investigation of whether to include observation prediction would be done in our future work.

---

> ### Author Response · Authors · 2022-11-18
> **Part I: Clarification on the Novelty of Random Masked Hindsight Control**
>
> We are grateful for the detailed feedback and very insightful suggestions by Reviewer CXdh. We are encouraged that Reviewer CXdh finds the method well-motivated, the experiments well-designed, and the paper well-written. We have added more ablation experiments as suggested by the reviewer, which can be found in Section 6.3 and Appendix B.5. Below we address all the questions and concerns raised by Reviewer CXdh.
>
> ---
>
> > **Q1:** The novelty of the proposed random masked hindsight control is a bit limited. Although the authors claimed the difference to BERT0like models, the general idea of mask-and-predict is the same and the only difference is the items to mask. This idea is also similar to those visual pre-training models like ViT etc.
>
> **A1:**
> We agree that the high-level idea of mask-and-predict is not new in language and vision training methods with a transformer backbone. But how to properly apply this idea to control tasks is not fully investigated by existing literature yet. We would like to emphasize that "which items to mask" and "how to define the prediction task" are very important and challenging. Even in the vision domain, how to design the best masking strategies is still an open problem studied by many recent papers[1,2]. In control problems, designing the masking strategy over state-action sequences can be more challenging given the environment uncertainty. Please note that the **ACL baseline** is exactly using a BERT-like loss, but SMART significantly outperforms ACL as shown in Figure 5, which suggests the advantages of our design.
>
> More importantly, the random masked hindsight control is a part of our control-centric pretraining objective. Our ablation study in Figure 6 suggests that **combining long-term control information and short-term control information** is a critical design for sequential decision making problems, which **distinguishes SMART** from other mask-and-predict methods.
>
> In summary, our main contribution is to formulate the pretraining-finetuning pipeline for control tasks, and to propose a pretraining paradigm, including the control-centric architecture and the pretraining objective, that is specifically designed for control tasks and significantly outperforms baseline methods.
>
>
> ---
>
> Refs:
>
> [1] Feichtenhofer, Christoph, et al. "Masked Autoencoders As Spatiotemporal Learners." NeurIPS 2022.
>
> [2] Tong, Zhan, et al. "Videomae: Masked autoencoders are data-efficient learners for self-supervised video pre-training." NeurIPS 2022.

---

### Official Review · Reviewer_EgV4 · 2022-10-27

**Confidence:** 3
**Clarity, Quality, Novelty And Reproducibility:** Good
**Correctness:** 4
**Technical Novelty And Significance:** 4
**Empirical Novelty And Significance:** 3
**Recommendation:** 8

**Strength And Weaknesses:**

Strengths:
+ The proposed method learns reward-agnostic representations in pre-training phase to improve generalizability
+ The proposed method effectively captures long-term dependency information using random-masked hindsight control mechanism.
+ Ablation studies demonstrates effectiveness of each component of control-centric objective function.
+ The paper is organized well and explained every aspect in clear terms.

Comments/Questions:
- Figure 1 finetuning phase shows that R_t is shown as input at all different time points: t, t+1 and t+L. Are there typos? Should it be R_{t+1} at t+1 and R_{t+L} at t+L time stamps?
- Proposed method was evaluated on DeepMind Control (DMC) suite tasks cheetah: run, cart-pole: swingup, cart-pole: balance, hopper: hop, hopper: stand, walker: stand, walker: run, walker: walk, pendulum: swingup, finger: spin. This dataset also contains other tasks such as ball-in-cup: catch, reacher: hard, finger: turn, manipulator: bring ball, swimmer: 6 links, swimmer: 15 links, fish: swim, cheetah: random action, hopper: random actions, walker: random actions, humanoid: random actions, humanoid: stand, humanoid: walk, humanoid: run, CMU motion capture. It would be interesting to see how the proposed method would perform on these slightly more challenging tasks.


**Summary Of The Paper:**

The manuscript proposes a new method which learns essential representations in a pretraining phase, including both long-term and short-term dependency information, in a sequential decision making process using reinforcement learning. For this, authors have proposed a novel control-centric objective which contains three components: forward dynamic prediction, inverse dynamics prediction, and random-masked hindsight control. First two components exploit local dependency information and the third component learns long-term dependency. The proposed method is first pretrained on multi-task dataset to learn reward agnostic representations which are utilized on downstream tasks. The method is evaluated on DeepMind Control (DMC) suite, demonstrating superior performance compared to scratch, single task pretrained, and ACL method reported by Yang et.al. 2021.

**Summary Of The Review:**

The proposed method is innovative and effective as demonstrated by their empirical study. The presentation of the paper is clear and code will be available to the research community to facilitate the reproduction of reported results and applications on other learning tasks.

---

> ### Author Response · Authors · 2022-11-18
> **We Added the Suggested Experiments**
>
> We greatly appreciate the detailed and insightful feedback of Reviewer EgV4. We are encouraged that Reviewer EgV4 finds our method innovative and effective. Below we address the questions of the reviewer in detail.
>
> > **Q1:** Figure 1 finetuning phase shows that R_t is shown as input at all different time points: t, t+1 and t+L. Are there typos? Should it be R_{t+1} at t+1 and R_{t+L} at t+L time stamps?
>
> **A1:** They are indeed typos. They should be $R_{t+1}$ and $t+1$ and $R_{t+L}$ at $t+L$. We have modified the figure in the revised paper. Thank you for pointing it out!
>
> ---
>
> > **Q2:** It would be interesting to see how the proposed method would perform on these slightly more challenging tasks.
>
> **A2:** Thank you for the suggestions! We have added more experimental **results on 5 more tasks** including ball-in-cup: catch, finger: turn-hard, fish: swim, swimmer: swimmer-6 links, and swimmer: swimmer-15 links. The results still match our previous conclusion in the paper and suggest that SMART generalizes well to these more challenging tasks. These new results and discussions are in Appendix B.1 (highlighted in blue). Here we still use the former pretrained checkpoint model which is trained on cartpole-swingup, hopper-hop, cheetah-run, walker-stand and walker-run, without any overlapping with the above 5 new tasks. Therefore, the fact that SMART still works well in these new tasks justifies its strong generalizability.
>
> It will be interesting to extend SMART to other tasks and settings. Since the experiment requires data collecting, pretraining and fintuning, it is hard to finish them all in the rebuttal period with limited computation resources. But we are happy to keep running the experiments and add the results to the camera-ready version. We will later release the code, the dataset and the pretrained models when the paper is de-anonymized.
>
>
> We also added more in-depth ablation studies in Section 6.3 and Appendix B.5 to further verify the effectiveness of our method. We also explored some variants of the current method as discussed in Appendix B.5.2. We hope these additional results can help establish a better understanding of our method and the pretraining-for-control problem.
>
> ---
>
> Thank you again for reviewing the paper. We hope our answers above and the additional experiments have addressed your questions. Please let us know if there are any other questions or concerns.

---

### Official Review · Reviewer_JbXg · 2022-11-03

**Confidence:** 2
**Correctness:** 4
**Technical Novelty And Significance:** 2
**Empirical Novelty And Significance:** 2
**Recommendation:** 6

**Clarity, Quality, Novelty And Reproducibility:**

This paper clearly presents the proposed methodology and the authors mention to release the code for reproducible research with camera ready version.



**Strength And Weaknesses:**

Strength: This paper proposes an effective approach for pertaining SSL model for multi-task sequential decision making. This paper clearly presents the proposed approach, with carefully designed extensive experiments on DeepMind Control Suite and comparison with a set of baselines.

Weakness: Overall, I find this paper has limited contribution. Also, the section 6.3 (Ablation and Discussion) is brief towards discussion and ablation studies can be extended.



**Summary Of The Paper:**

In this paper, authors present a method for pretraining a generalisable and resilient SSL model with control transformer for multi-task sequential decision making. They propose a self-supervised and control-centric objective that encourages the transformer-based model to capture control-relevant representation. The evaluation is performed on multiple domains and tasks. The results show model’s effectiveness and robustness to distribution shift.



**Summary Of The Review:**

The proposed SMART method for training SSL model with control transformer for multi-task sequential decision making is shown as effective and robust. This paper clearly presents the proposed approach and conducts extensive experiments on multiple tasks and comparison with multiple baselines.

I think authors should consider improving section 6.3 with more discussion on the results of ablation experiments.

Additonally, I find this paper has limited contribution with control transformers.

---

> ### Author Response · Authors · 2022-11-18
> **Part II: Clarification on Contribution and Novelty**
>
> > **Q2.** Additonally, I find this paper has limited contribution with control transformers.
>
> **A2:** We would like to emphasize the novelty and contributions of this paper. Our main contribution lies in two folds:
>
> **1. Novelty and Contribution in Problem Setup and Pretraining Paradigm.**
> We comprehensively studied previous pretraining works for RL and formulate the pretraining-finetuning pipeline for control tasks, which has not been well-studied in previous work. This paper formulates the multi-task pretraining and finetuning pipeline for control tasks (Sec. 4). As discussed in Sec. 2 and Sec. 4, most previous pretraining methods for RL are designed for a single task, or require expert supervision for pretraining. In contrast, our formulation of pretraining-finetuning pipeline is more generic and flexible. Upon this, we then propose an effective pretraining approach SMART, which pretrains a single representation model that can efficiently adapt to multiple tasks (dynamics and rewards) across different domains (state-action spaces). On the other hand, SMART focuses on self-supervised representation pretraining, which is different from generalist agents (e.g. GATO[1]) which directly learn policies for multiple tasks with a single model. Therefore, our method addresses a rarely studied but important problem, and significantly differs from existing work.
>
> **2. Novelty and Contribution in Design of Architecture and Objective.**
> SMART shows good versatility, generalizability and resillience, thanks to the design of the control transformer and the self-supervised control-centric pretraining objective. Our experiments and ablation studies suggest the following messages.
> - **[Architecture]** Control transformer does not require reward signals, and thus is more versatile for various learning scenarios (with or without reward), and more general for multiple tasks with different reward setups.
> - **[Objective]** (1) Both short-term control information and long-term control information are important for pretraining in sequential decision making tasks (verified by ablation study). (2) Self-supervised pretraining reduces overfitting to the pretraining behaviors, and thus makes the model more resilient to low-quality behavior policy used in pretraining data (verified by the comparison to decision transformer).
>
> ---
>
> We greatly appreciate your time and suggestions. We hope the above explanations and the additional experiments have addressed your concerns and questions. Please let us know if there are further questions.
>
>
> ---
> Refs:
>
> [1] Reed, Scott, et al. "A generalist agent."

---

> ### Author Response · Authors · 2022-11-18
> **Part I: More Extensive Ablation Studies and Discussions**
>
> We thank Reviewer JbXg for the valuable feedback. We are encouraged that Reviewer JbXg finds our method effective and robust. Below we address the concerns raised by the reviewer.
>
> > **Q1.** Weakness: Overall, I find this paper has limited contribution. Also, the section 6.3 (Ablation and Discussion) is brief towards discussion and ablation studies can be extended.
>
> **A1:** Thank you for the suggestion. Due to the space limit, we are sorry that the original manuscript puts many interesting results and ablation studies in the Appendix. We will move them to the main paper in a longer version of this paper. These ablation studies and discussions include:
>
> 1. how SMART works when one of the three terms in the pretraining objective is removed (Sec. 6.3);
> 2. whether including reward information in the pretraining helps or not (Sec. 6.3);
> 3. how sensitive SMART is to the model capacity, including both model depth (number of Transformer blocks) and model width (representation dimension) (Sec. 6.3);
> 4. how a transformer backbone is different from a ResNet backbone used in some prior work (App. B.3);
> 5. how SMART can be incorporated into online reinforcement learning (App. B.4).
>
> In the revised paper, we have conducted **additional experiments and broader ablation studies**, including
>
> 6. results of pretraining with only short-term control information (forward + inverse) or only long-term control information (random masked hindsight control), verifying the importance of both perspectives (Sec. 6.3 + App. B.5.1);
> 7. several variants of the objective selection (e.g., using a fixed mask size, adding a contrastive loss, etc.), which shows the effectiveness of our design, as well as the flexibility of the method (App. B.5.2);
> 8. additional results of using smaller-size models (App. B.2)
> 9. additional results on 5 more unseen tasks with larger discrepancy from pretraining tasks, to verify the generalizability of SMART (App. B.1).
>
>
> In summary, we have provided a large amount of empirical results to discuss the effectiveness and potential extensions of our method. Please note that our experiments include both pretraining and finetuning in multiple tasks, which are on a large scale. There could be combinatorially many variants and extensions of the methods, while it is prohibitive to test all of them under limited time and resources. Therefore, it is more important to get insights from these experiments. We believe that the experimental efforts and discussions in this paper are already extensive and sufficient to (1) demonstrate the versatility, generalizability and resilience of SMART, (2) verify the effectiveness of the algorithmic design of SMART, and (3) provide insights on future extension and application of SMART.

---

> ### Author Response · Authors · 2022-12-12
> **Does our response address your concerns? We are happy to address any further questions.**
>
> Dear Reviewer JbXg,
>
> We would like to first thank you again for your constructive comments and helpful suggestions. Since we are nearly at the end of the discussion phase, we hope to discuss further with you whether your concerns have been addressed by our previous reponse and additional experiments. If you still have any questions regarding our work, please let us know. We are very happy to discuss further and improve the work.
>
> Best,
>
> Paper4962 Authors

---

### Author Response · Authors · 2022-11-18
**Geneal Response and Summary of Paper Updates**

We greatly appreciate the insightful and high-quality comments from all reviewers. We are particularly encouraged that all reviewers give high ratings to our paper and recognize the effectiveness of the proposed method.

---

We have addressed all questions of reviewers in each individual response. According to the suggestions of reviewers, we also conduct a series of additional experiments and include the results in the revised paper. The modifications in the paper are highlighted in blue. Below is a summary of paper updates and new experiments.

- We provide more thorough and in-depth **ablation studies** and discussions of the proposed pretraining objective, in Section 6.3 and Appendix B.5.1. The results verify the importance of both short-term control information and long-term control information.
- We evaluate several **variants** of SMART in Appendix B.5.2, which modifies the objective or adds new components to the objective. The results show the flexibility and effectiveness of the proposed design.
- We further show how the pretrained SMART model works on **5 more tasks** that have a larger discrepancy with pretraining tasks. The results provided in Appendix B.1 demonstrate the strong generalizability of SMART.
- In addition to the **model capacity** experiments in Figure 7(b), we further vary the model depth and present the results in Appendix B.2.

We hope the updated results and our responses have addressed the questions that reviewers have. Please let us know if there are any other questions or suggestions.

Thanks,

Paper4962 Authors

---

### Decision · Program_Chairs · 2023-01-20

**Decision:**

Accept: notable-top-25%

**Justification For Why Not Higher Score:**

This is a very well-done paper, but experiments were limited to continuous control tasks, which hinders its broad applicability.

**Justification For Why Not Lower Score:**

All reviewer concerns were addressed as far as I can see, and I’m confident there will be a lot of interest in this paper at the conference. I believe it warrants a spotlight for that reason.

**Metareview: Summary, Strengths And Weaknesses:**

This paper presents a new method for pretraining essential representations for fine tuning on downstream sequential decision-making tasks using a Control Transformer and self-supervised learning.

All reviewers agreed on the importance of this topic, the clarity of presentation, and the thoroughness and quality of the experiments. A few clarification issues were raised and some further control experiments were suggested in the reviews. Authors were very responsive during the rebuttal period, including adding results on 5 more tasks, adding an experiment with observation prediction and various other variants of their model, comprehensive ablations. Their results show that it’s important to learn both short-term and long-term dependencies for useful representations, which should be of interest to all ICLR attendees interested in deep RL and effectiveness of pretraining/fine tuning.

Although 3 out of 4 reviewers failed to respond to author rebuttals, after reading through everything, I'm satisfied that authors have adequately addressed their critiques. I’m happy to recommend acceptance, and look forward to seeing the updates and the release of the code with the camera ready.


**Note From Pc:**

if the above contains the word "oral" or "spotlight" please see: "oral" presentation means -> notable-top-5% and "spotlight" means -> notable-top-25%. As stated in our emails, we are disassociating presentation type from AC recommendations